# Massive cortical reorganization in sighted Braille readers

**Katarzyna Siuda-Krzywicka[1,2†], Łukasz Bola[1,3†], Małgorzata Paplińska[4], Ewa Sumera[5], Katarzyna Jednoróg[6], Artur Marchewka[3], Magdalena W Śliwińska[7], Amir Amedi[8,9,10,11], Marcin Szwed[1]***

[1]Department of Psychology, Jagiellonian University, Kraków, Poland; [2]INSERM U 1127, CNRS UMR 7225, Sorbonne Universités, and Université Pierre et Marie Curie-Paris 6, UMR S 1127, Institut du Cerveau et de la Moelle épinière (ICM), Paris, France; [3]Laboratory of Brain Imaging, Neurobiology Center, Nencki Institute of Experimental Biology, Warsaw, Poland; [4]Academy of Special Education in Warsaw, Warsaw, Poland; [5]Institute for the Blind and Partially Sighted Children in Krakow, Kraków, Poland; [6]Laboratory of Psychophysiology, Department of Neurophysiology, Nencki Institute of Experimental Biology, Warsaw, Poland; [7]Department of Experimental Psychology, University College London, London, United Kingdom; [8]The Cognitive Science Program, The Hebrew University of Jerusalem, Jerusalem, Israel; [9]Department of Medical Neurobiology, The Institute for Medical Research Israel-Canada, Faculty of Medicine, The Hebrew University of Jerusalem, Jerusalem, Israel; [10]The Edmond and Lily Safra Center for Brain Sciences, The Hebrew University of Jerusalem, Jerusalem, Israel; [11]Sorbonne Universite´s, UPMC Univ Paris 06, Institut de la Vision, Paris, France

**\*For correspondence:** mfszwed@gmail.com

[†]These authors contributed equally to this work

**Competing interests:** The authors declare that no competing interests exist.

**Abstract** The brain is capable of large-scale reorganization in blindness or after massive injury. Such reorganization crosses the division into separate sensory cortices (visual, somatosensory...). As its result, the visual cortex of the blind becomes active during tactile Braille reading. Although the possibility of such reorganization in the normal, adult brain has been raised, definitive evidence has been lacking. Here, we demonstrate such extensive reorganization in normal, sighted adults who learned Braille while their brain activity was investigated with fMRI and transcranial magnetic stimulation (TMS). Subjects showed enhanced activity for tactile reading in the visual cortex, including the visual word form area (VWFA) that was modulated by their Braille reading speed and strengthened resting-state connectivity between visual and somatosensory cortices. Moreover, TMS disruption of VWFA activity decreased their tactile reading accuracy. Our results indicate that large-scale reorganization is a viable mechanism recruited when learning complex skills.

## Introduction

The current view of neural plasticity in the adult brain sees it as ubiquitous but generally constrained by functional boundaries (*Hoffman and Logothetis, 2009*). At the systems level, experience-driven plasticity is thought to operate within the limits of sensory divisions, where the visual cortex processes visual stimuli and responds to visual training, the tactile cortex processes tactile stimuli and responds to tactile training, and so on. A departure from this rule is usually reported only during large-scale reorganization induced by sensory loss or injury (*Hirsch et al., 2015*; *Lomber et al., 2011*; *Merabet and Pascual-Leone, 2010*; *Pavani and Roder, 2012*). The ventral visual cortex, in particular, is activated in blind subjects who read Braille (*Büchel et al., 1998*; *Reich et al., 2011*;

**eLife digest** According to most textbooks, our brain is divided into separate areas that are dedicated to specific senses. We have a visual cortex for vision, a tactile cortex for touch, and so on. However, researchers suspect that this division might not be as fixed as the textbooks say. For example, blind people can switch their 'leftover' visual cortex to non-visual purposes, such as reading Braille – a tactile alphabet.

Can this switch in functional organization also happen in healthy people with normal vision? To investigate this, Siuda-Krzywicka, Bola et al. taught a group of healthy, sighted people to read Braille by touch, and monitored the changes in brain activity that this caused using a technique called functional magnetic resonance imaging. According to textbooks, tactile reading should engage the tactile cortex. Yet, the experiment revealed that the brain activity critical for reading Braille by touch did not occur in the volunteers' tactile cortex, but in their visual cortex.

Further experiments used a technique called transcranial magnetic stimulation to suppress the activity of the visual cortex of the volunteers. This impaired their ability to read Braille by touch. This is a clear-cut proof that sighted adults can re-program their visual cortex for non-visual, tactile purposes.

These results show that intensive training in a complex task can overcome the sensory division-of-labor of our brain. This indicates that our brain is much more flexible than previously thought, and that such flexibility might occur when we learn everyday, complex skills such as driving a car or playing a musical instrument.

The next question that follows from this work is: what enables the brain's activity to change after learning to read Braille? To understand this, Siuda-Krzywicka, Bola et al. are currently exploring how the physical structure of the brain changes as a result of a person acquiring the ability to read Braille by touch.

*Sadato et al., 2002*, *1996*) and lesions of this area impair Braille reading (*Hamilton et al., 2000*). This visual cortex thus has the innate connectivity required to carry out a complex perceptual task – reading – in a modality different than vision. It is unclear, however, to what extent this connectivity is preserved and functional in the normal, adult brain.

A growing amount of evidence suggests that some form of cross-modal recruitment of the visual cortex could be possible in the normal healthy adults (*Amedi et al., 2007*; *Kim and Zatorre, 2011*; *Powers et al., 2012*; *Saito et al., 2006*; *Zangenehpour and Zatorre, 2010*; *Merabet et al., 2004*; *Zangaladze et al., 1999*). Nonetheless, the behavioural relevance of these cortical mechanisms remains unclear, especially for complex stimuli. Notably, several experiments failed to find such cross-modal reorganization in sighted subjects, even after extensive training (*Kupers et al., 2006*; *Ptito et al., 2005*). One study found cross-modal plastic changes in subjects that were blindfolded for several days (*Merabet et al., 2008*), but these plastic changes quickly vanished once the subjects removed their blindfolds.

Here, for the first time, we show that large-scale, cross-modal cortical reorganization is a viable, adaptive mechanism in the sighted, adult brain. In our experiment, sighted adults followed a 9-month Braille reading course. The resulting cortical changes were tracked using task-based and resting-state functional Magnetic Resonance Imaging (fMRI) and manipulated with Transcranial Magnetic Stimulation (TMS).

## Results

Twenty-nine sighted adults, Braille teachers and professionals, students specializing in the education of the blind, and family members of blind people, took part in the study. Some subjects were familiar with Braille signs visually (some teachers actually knew how to read visual Braille upside-down, as upside-down was the usual orientation at which they viewed their students' work laid out on school benches). All of them were naive in tactile Braille reading (Appendix 1.1). All the participants completed an intensive, tailor-made, 9-month-long tactile Braille reading course (Materials and methods). The majority progressed significantly in tactile reading, reaching an average performance of

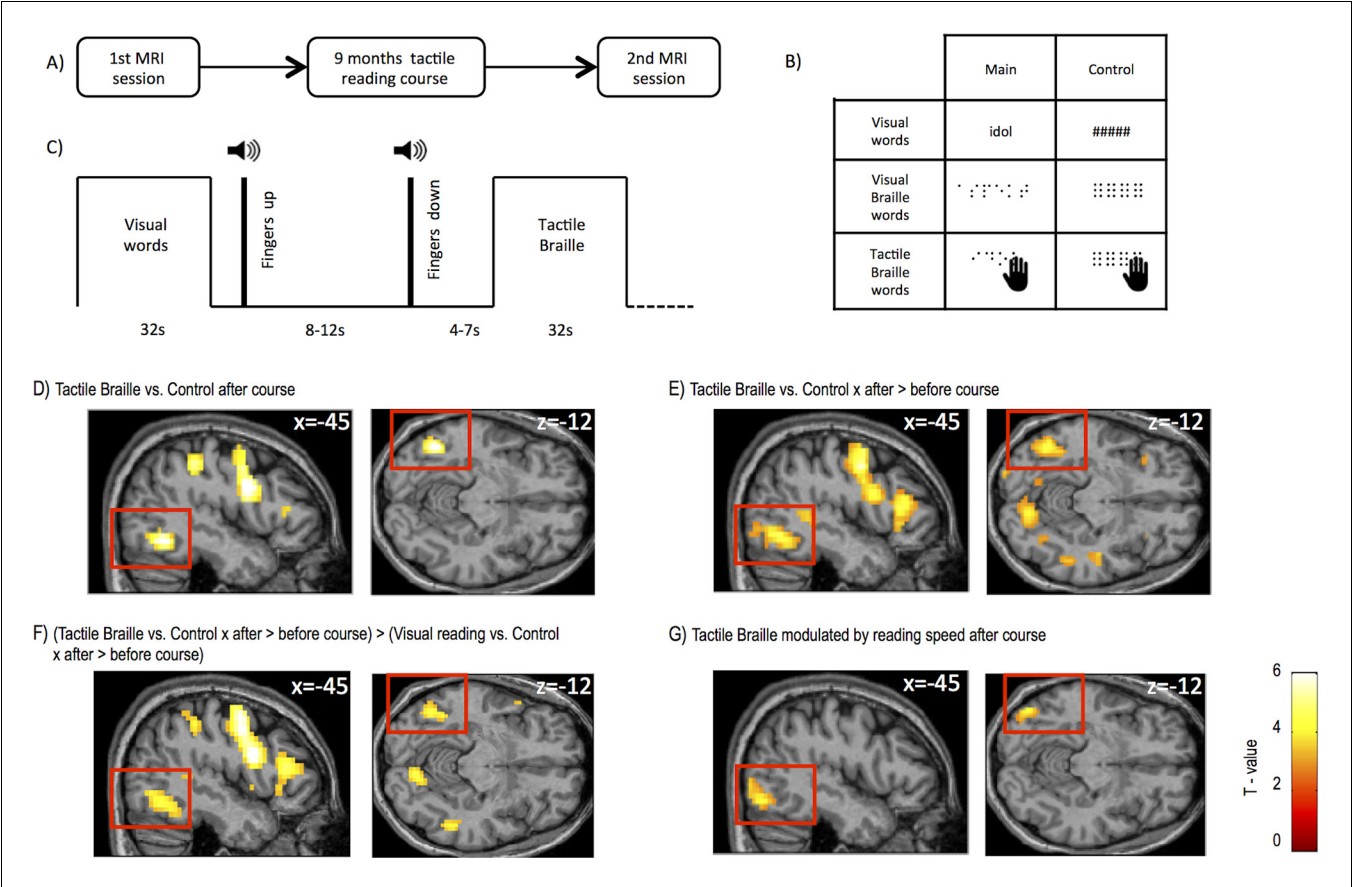

**Figure 1.** Experimental design and brain regions activated by tactile Braille reading. (**A**) The study consisted of two identical MRI sessions performed before and after an intensive 9-month tactile Braille course. In the fMRI experiment (**B**), subjects viewed visual words written in the regular alphabet and Braille words displayed on a screen (visual Braille) and touched tactile Braille words. As a control, they viewed strings of hash signs and meaningless pseudo-Braille dots and touched meaningless pseudo-Braille dots. The experiment used a block design (**C**); after each block, the subjects lifted their fingers from the table on which the tactile stimuli were presented. A new board with tactile stimuli was then placed on the table, the subjects put their fingers down, and a new block began. (**D**) Compared to the tactile control, tactile reading after the course evoked activations in the visual word form area (VWFA), the lateral occipital area and other areas (*Tables 1–3*). Similar areas were activated when we computed (**E**) an interaction between tactile Braille/tactile control stimuli and the before- and after-course time points and for (**F**) an interaction between tactile Braille/tactile control vs. visual reading/visual control stimuli and the before- and after-course time points. The latter confirmed that the increase in visual cortex activation after the course was specific for tactile reading. (**G**) When we modeled the modulation of fMRI responses to tactile Braille by the subjects' reading speed, the only significant whole-brain correlate with tactile reading proficiency was found in the ventral visual system (k=103 voxels). Voxel-wise thresholds: (**D**) p<0.001; (**E–G**) p<0.005; Cluster-wise thresholds: (**D–F**) p<0.05; (**G**) uncorrected, k=100 voxels. For control fMRI experiment, see *Figure 1—figure supplement 1* for procedures and *Figure 1—figure supplement 2* for results. For supplementary fMRI results, see *Figure 1—figure supplement 3*.

The following figure supplements are available for figure 1:

**Figure supplement 1.** Control experiment stimuli and procedures.

**Figure supplement 2.** Main results of the control experiment.

**Figure supplement 3.** Supplementary fMRI results

6.20 words per minute read aloud (WPM) (SD=3.94, range = 0–17, Appendix 1.1) at the end of the course. Because the course curriculum included learning to recognize Braille letters by sight, the subjects also improved in reading Braille characters visually (Appendix 1.2).

Before and after the tactile Braille reading course, the subjects took part in an fMRI experiment (*Figure 1A–C*; 'Materials and methods'), in which they viewed regular visual words, Braille words displayed visually on a screen (visual Braille words) and touched tactile Braille words. Additional control

**Table 1.** Summary of main activations for tactile Braille reading contrasted with control conditions and visual reading across the tactile Braille course

| Contrast | Voxel-wise p threshold | Region | BA | Hemisphere | Z score | Cluster size | MNI coordinates | | |
|---|---|---|---|---|---|---|---|---|---|
| Tactile Braille vs. Control after training | p = 0.001 | Fusiform Gyrus (VWFA) | 37 | Left | 5.18 | 75 | -45 | -58 | -12 |
| | | Inferior Frontal Gyrus | 9 | Left | 5.15 | 467 | -42 | 2 | 27 |
| | | | 46 | Left | 3.43 | 97 | -42 | 32 | 11 |
| | | | 47 | Right | 4.48 | 111 | 33 | 26 | 3 |
| | | | 45 | Right | 3.82 | 66 | 60 | 11 | 23 |
| | | Medial Frontal Gyrus | 6 | Left | 4.84 | 467 | -6 | 2 | 63 |
| | | Superior Occipital Gyrus | 39 | Right | 5.08 | 208 | 33 | -73 | 27 |
| | | Precuneus | 7 | Right | 4.63 | 208 | 21 | -70 | 47 |
| | | | 7 | Left | 5.01 | 401 | -21 | -73 | 43 |
| | | Inferior Parietal Lobule | 40 | Left | 4.55 | 401 | -30 | -43 | 43 |
| | | | 40 | Right | 4.24 | 93 | 45 | -34 | 47 |
| | | Middle Frontal Gyrus | 46 | Right | 3.55 | 111 | 42 | 29 | 19 |
| | | Cerebellum | * | Right | 4.36 | 274 | 27 | -64 | -24 |
| | | Insula | 13 | Left | 4.34 | 97 | -33 | 23 | 3 |
| Tactile Braille vs. Control x after > before training | p = 0.005 | Superior Temporal Gyrus | 22 | Right | 5.49 | 225 | 48 | -25 | -1 |
| | | Fusiform Gyrus (VWFA) | 37 | Left | 4.54 | 606 | -45 | -61 | -12 |
| | | Middle Frontal Gyrus | 6 | Left | 4.48 | 1031 | -45 | -1 | 43 |
| | | Inferior Frontal Gyrus | 45 | Left | 4.22 | 1031 | -33 | 26 | 7 |
| | | | 47 | Right | 4.28 | 339 | 39 | 26 | -1 |
| | | Insula | 13 | Right | 3.95 | | 33 | 23 | 7 |
| | | Middle Temporal Gyrus | 22 | Left | 4.13 | 183 | -51 | -43 | 7 |
| | | | 37 | Left | 3.92 | 606 | -45 | -64 | -9 |
| | | Superior Temporal Gyrus | 22 | Left | 3.62 | 183 | -63 | -40 | 11 |
| | | | 22 | Left | 3.43 | | -57 | -52 | 11 |
| | | Precuneus | 7 | Left | 4.10 | 606 | -18 | -67 | 43 |
| | | | 7 | Right | 3.78 | 713 | 24 | -73 | 31 |
| | | Cerebellum | * | Right | 4.01 | 713 | 27 | -64 | -28 |
| | | Middle Occipital Gyrus | 37 | Right | 3.85 | 713 | 39 | -70 | -5 |
| Tactile Braille vs.Control – Visual Reading vs.Control x after > before training | p = 0.005 | Inferior Frontal Gyrus | 9 | Left | 5.60 | 2469 | -45 | 5 | 23 |
| | | Superior Frontal Gyrus | 6 | Left | 5.48 | | -3 | 5 | 63 |
| | | Precentral Gyrus | 6 | Left | 5.39 | | -45 | -4 | 51 |
| | | Cerebellum | * | Right | 4.78 | 1154 | 6 | -79 | -36 |
| | | | * | Left | 3.49 | | -9 | -82 | -20 |
| | | Fusiform Gyrus (VWFA) | 37 | Left | 4.05 | 1154 | -45 | -61 | -12 |
| | | Inferior Temporal Gyrus | 37 | Left | 3.81 | | -48 | -70 | -5 |
| | | Inferior Parietal Lobule | 40 | Left | 4.21 | 532 | -33 | -49 | 51 |
| | | Precuneus | 19 | Left | 4.08 | | -27 | -73 | 31 |
| | | | 7 | Right | 4.31 | 738 | 27 | -70 | 31 |
| | | Cuneus | 30 | Right | 4.02 | | 27 | -76 | 3 |

conditions included visual strings of hash symbols (control for visual words), visual strings of pseudo-Braille characters (control for visual Braille), and tactile strings of pseudo-Braille characters (control for tactile Braille) (*Figure 1B* and 'Materials and methods'; a pseudo-Braille character contains six dots and has no meaning). The subjects could not see any of the tactile stimuli. Subjects also performed a control mental imagery fMRI experiment ('Materials and methods', *Figure 1—figure supplements 1* and *2*, Appendix 1.5).

## Whole-brain fMRI analysis

Whole-brain fMRI analysis showed that after the course, activation to tactile Braille reading relative to the tactile control condition peaked within the Visual Word Form Area (VWFA), (*Figure 1D*, peak MNI = -45 -58 -12, Z = 5.18), a ventral visual stream (*Van Essen, 2005*) region known to be involved in visual reading (*Dehaene and Cohen, 2011*; *Price and Devlin, 2011*; *Szwed et al., 2011*). Additional activations were observed in the intraparietal sulcus, supramarginal gyrus, and frontal areas (*Table 1*). We observed the same increase in VWFA activation when we contrasted tactile Braille activation before and after the course [tactile Braille vs. tactile control x (after course-before course)] (*Figure 1E*, peak: MNI = -45 -64 -12, Z=5.02). This contrast also revealed clusters in the middle occipital gyrus, cuneus (BA18), the left superior temporal gyrus, and frontal areas (*Table 1*). To determine whether these changes were due to a general activity increase for all reading conditions, we computed a whole-brain ANOVA interaction analysis of the signal change following the course for reading visual words, visual Braille words and tactile Braille words versus their respective control tasks ([tactile Braille vs. tactile control - (visual Braille vs. visual Braille control+visual words vs. visual words control)] x (after course-before course)) (see 'Materials and methods'). Relative to visual reading, tactile reading led to increased activation in the VWFA (*Figure 1F*; MNI -45 -61 -12, Z = 4.05) and other parietal and frontal areas (*Table 1*).

The results demonstrated that this pattern of activation in visual cortex was observed specifically for tactile reading. Indeed, we found no increase in activation to visual words following the course. The only increases in activation to visual Braille were found in the default mode network nodes in the parietal and prefrontal cortices (*Figure 1—figure supplement 3C*, *Table 2*, Appendix 1.3). Importantly, the analyses mentioned in this section (*Figure 1D–F*), including the whole-brain ANOVA, did not reveal any activation in the primary and secondary somatosensory cortices, even at an exploratory threshold of p=0.01 voxel-wise.

## fMRI – subject behavior correlations

Our subjects' progress in tactile reading was not homogeneous, with some subjects being entirely unable to learn Braille at all (0 WPM) and other subjects reaching a speed of 17 WPM (Appendix 1.1). We therefore used regression to ask which fMRI responses to tactile Braille reading were modulated by the subjects' tactile reading speed. This regression analysis revealed one significant cluster, located in the left inferior occipital gyrus (*Figure 1G*; MNI = -45 -76 -12, Z = 3.69, *Table 3*). A similar result was obtained when we correlated single letter recognition speed with tactile Braille activations, which further supports the significance of the visual cortex in learning to read Braille (*Table 3*, Appendix 1.4). We found no such correlations for other reading speed measures (e.g. visual Braille reading speed) or imagery activations (e.g. imagining tactile Braille; see Appendix 1.5). These observations indicate that the ventral visual activations for tactile reading cannot be explained as a by-product of imagery.

## Representation similarity analysis

Visual words, visual Braille and tactile Braille all elicited activity in the VWFA. How similar are the neural representations of these three scripts? The similarity of neural representations can be studied with multivariate pattern analysis-representational similarity analysis (MVPA-RSA). This method is based on the premise that stimuli that share similar neural representations will generate similar voxel activation patterns (*Kriegeskorte et al., 2008*; *Rothlein and Rapp, 2014*). Using MVPA-RSA ('Materials and methods'), we found that the two Braille conditions (visual and tactile) had the most similar activation patterns in the VWFA (*Figure 2A,r*=0.48), despite very large differences in the magnitude of the two activations (see *Figure 2B*). The correlations of the two Braille conditions with visual words were much weaker (*Figure 2A*, 'Materials and methods' and Appendix 1.6). Despite a

**Table 2.** Summary of main activations for visual reading and visual Braille reading contrasted with control conditions across the tactile Braille course.

| Contrast | Voxel-wise p treshold | Region | BA | Hemisphere | Z score | Cluster size | MNI coordinates | | |
|---|---|---|---|---|---|---|---|---|---|
| Visual words vs. Control before training | p = 0.005 | Inferior Temporal Gyrus | 37 | Left | 7.43 | 359 | -42 | -70 | -9 |
| | | Fusiform Gyrus (VWFA) | 37 | Left | 5.02 | | -42 | -52 | -16 |
| | | Middle Occipital Gyrus | 18 | Left | 3.17 | | -24 | -94 | 3 |
| Visual Braille vs Control before training | p = 0.005 | Precuneus | 19 | Right | 5.83 | 1767 | 30 | -64 | 39 |
| | | Middle Temporal Gyrus | 37 | Right | 5.58 | | 45 | -64 | -9 |
| | | Middle Occipital Gyrus | 19 | Right | 5.19 | | 48 | -76 | -1 |
| | | | 19 | Left | 5.60 | 1640 | -45 | -85 | -1 |
| | | Inferior Frontal Gyrus | 9 | Left | 5.81 | 1374 | -42 | 2 | 27 |
| | | Middle Frontal Gyrus | 6 | Left | 5.20 | | -30 | -4 | 55 |
| | | Precentral Gyrus | 6 | Left | 5.11 | | -51 | -4 | 39 |
| | | Fusiform Gyrus (VWFA) | 37 | Left | 5.62 | 1640 | -45 | -58 | -12 |
| | | Inferior Occipital Gyrus | 19 | Left | 5.59 | | -42 | -73 | -9 |
| Visual Braille vs. Control x after > before training | p = 0.005 | Precuneus | 7 | Right | 4.88 | 797 | 6 | -64 | 35 |
| | | | 31 | Left | 4.29 | | -9 | -61 | 27 |
| | | Middle Temporal Gyrus | 39 | Right | 4.04 | 275 | 51 | -64 | 23 |
| | | | 39 | Left | 4.00 | 297 | -45 | -67 | 27 |
| | | | 39 | Left | 3.86 | | -54 | -70 | 23 |
| | | Superior Temporal Gyrus | 13 | Right | 3.83 | 275 | 45 | -49 | 23 |
| | | Inferior Parietal Lobule | 40 | Right | 2.78 | | 60 | -52 | 43 |
| | | Superior Parietal Lobule | 7 | Left | 3.45 | 297 | -45 | -67 | 51 |
| | | Medial Frontal Gyrus | 10 | Left | 3.79 | 1184 | -3 | 62 | -1 |
| | | | 10 | Left | 3.73 | | -9 | 50 | -5 |
| | | Superior Frontal Gyrus | 8 | Right | 3.64 | | 9 | 47 | 43 |

difference in modality (visual vs. tactile), the neural representations of the two versions of Braille script were partially similar and distinct from the well-established representation for visual words.

## Region-of-interest (ROI) analysis

A region-of-interest (ROI) analysis was then applied (see 'Materials and methods'). In the VWFA (*Figure 2B*; all ROIs are in the left hemisphere) following the Braille course, the response to tactile Braille words changed from de-activation to positive activation, resulting in a significant difference between tactile words and their control (interaction: F(1,28)=18.5; p<001). This emerging difference was also driven by a decrease in activation to control tactile stimuli. Post-hoc t-tests (one for tactile words and another for tactile control, before vs. after course) showed that the two effects were of similar magnitude and neither of them reached statistical significance on their own. The VWFA also showed strong responses to visual Braille words, similar to previously reported responses to novel visual alphabets (e.g. *Szwed et al., 2014*; *Vogel et al., 2014*). These responses remained unchanged throughout the course. The lateral occipital area (*Figure 2C*) showed a similar emergence of responses to tactile Braille words after the course as well.

Several experiments have shown changes in primary somatosensory (SI) cortex activation to tactile stimuli after tactile training, in both humans (e.g. *Pleger et al., 2003*) and rodents (e.g. *Guic et al.,*

**Table 3.** Summary of activations for behavioural measures modulating the activity in reading.

| Contrast | Voxel-wise p treshold | Region | BA | Hemisphere | Z score | Cluster size | MNI coordinates | | |
|---|---|---|---|---|---|---|---|---|---|
| Tactile reading speed x activations in Tactile Braille after training | p = 0.005 | Inferior Occipital Gyus | 19 | Left | 3.69 | 103 | -45 | -76 | -13 |
| | | Middle Occipital Gyrus | 19 | Left | 3.65 | 103 | -39 | -85 | -1 |
| Tactile letter recognition x activations in Tactile Braille after the training | p = 0.005 | Middle Occipital Gyrus | 19 | Left | 3.62 | 170 | -48 | -73 | -12 |
| | | | 19 | Left | 3.21 | 170 | -39 | -85 | -1 |
| | | Lingual Gyrus | 18 | Left | 3.42 | 170 | -15 | -85 | -16 |
| | | Fusiform Gyrus (VWFA) | 19 | Left | 3.17 | 170 | -36 | -82 | -20 |
| | | | 37 | Right | 3.17 | 63 | 36 | -49 | -20 |
| | | Inferior Occipital Gyrus | 18 | Left | 3.04 | 170 | -39 | -85 | -16 |
| | | Cerebellum | * | Right | 3.00 | 63 | 30 | -58 | -12 |
| | | Cuneus | 17 | Right | 3.01 | 44 | 18 | -85 | 7 |
| | | | 18 | Right | 2.76 | 44 | 6 | -91 | 7 |

*2008*). In the current study, however, there was no significant change in activation to tactile Braille words and no after-course differences between tactile Braille words and the tactile control (*Figure 2D*). In the secondary somatosensory cortex (MNI: -51 -25 15), we found only a non-specific drop in activation to all tactile stimuli (F(1,28)=7.62, p=0.01), with no difference between tactile Braille words and the tactile control after the course. A drop in activation for the control condition was observed in the posterior attentional network (intraparietal sulcus, IPS, *Figure 2E*, t(28)=2.76, p=0.01). Those activation drops in the somatosensory cortices and in IPS are the most likely cause behind the activation drop for control tactile stimuli observed in the VWFA and LO (*Figure 2B–C*). In the primary motor cortex (MNI: -39 -25 59), we observed no such drop, which suggests that the finger-movement patterns across sessions and conditions remained unchanged. Thus, the Braille reading course led to an increase in activation to tactile Braille words in visual areas but not in the somatosensory cortex.

## Effects on resting-state fMRI

Learning can impact the resting-state activity of the brain (e.g. *Lewis et al., 2009*). Following the course, we observed an increase in resting-state functional connectivity between the VWFA seed and the left primary somatosensory cortex (*Figure 3A*, red; p<0.001, corrected for multiple comparisons; Z=4.57; MNI: -57 -24 45; 'Materials and methods'). This increase was the only statistically significant positive effect found. The same comparison showed a decrease in the VWFA's functional connectivity with other visual areas, bilaterally. Furthermore, after the course, the VWFA-S1 functional connectivity level was correlated with the subjects' progress in tactile Braille reading speed (in the month preceding the after-course scan, *Figure 3*; r(27)=0.49, p=0.007). The VWFA thus increased its coupling with the somatosensory cortex while decreasing its coupling with other visual areas. This VWFA-S1 functional connectivity was behaviorally relevant for tactile reading and was likely to be dynamically modulated in a relatively short learning period (similar to: *Lewis et al., 2009*) (see also Appendix 1.7).

## Transcranial magnetic stimulation

Finally, to test the role of the VWFA in Braille reading, we performed a repetitive Transcranial Magnetic Stimulation (rTMS) experiment in which nine subjects were tested after the course in tactile Braille reading (reading speeds: 3–17 WPM). rTMS was applied to the VWFA and to two control sites – the lateral occipital area and the vertex. Both the VWFA and the lateral occipital area were

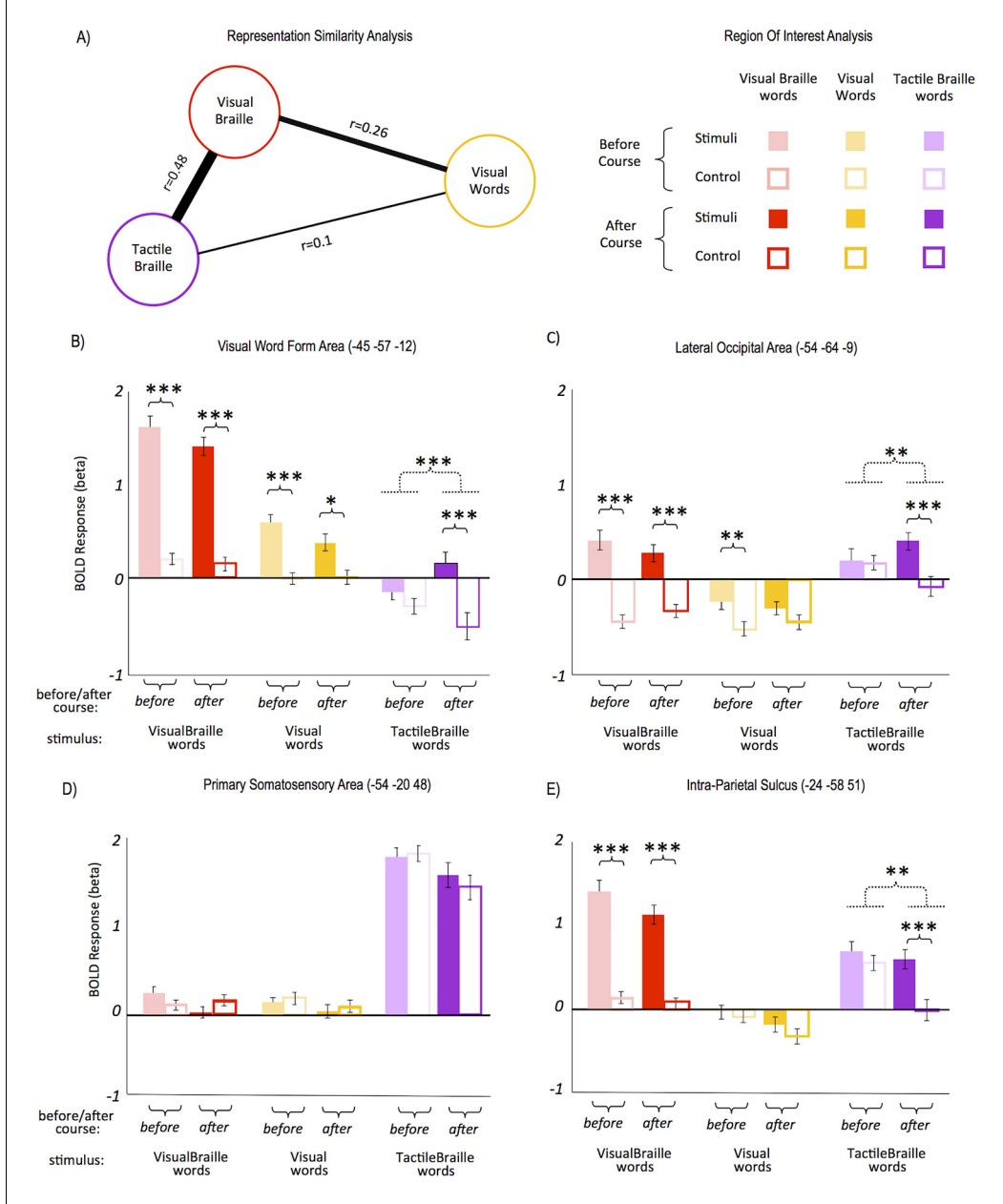

**Figure 2.** Response similarity and region-of-interest (ROI) analyses. Response similarity analysis showed that (**A**) the activity patterns for both Braille alphabets were the most similar, whereas the patterns for tactile Braille and visual words differed the most. In the VWFA (**B**), the response to tactile Braille words changed from a de-activation to a positive activation. The VWFA also showed strong responses to visual Braille words; these, however, did not change significantly following the Braille course. The lateral occipital area (**C**) also saw the emergence of responses to tactile Braille words similar to the VWFA. In contrast, there was no effect of the Braille course in the somatosensory cortex (**D**). A drop in activation for the control condition was salient in the intraparietal sulcus (**F**), in which the activation to tactile Braille words remained unchanged, whereas the activation to the tactile control dropped to zero. Arrow thickness and the distance between scripts in (**A**) are proportional to correlation strength. (***) p<0.001; (**) p<0.01; (*) p<0.05. Dashed lines denote interactions. All ROIs are in the left hemisphere.

localized using the individual subjects' fMRI results ('Materials and methods'). Similar to a previous visual reading study (*Duncan et al., 2010*), we chose the lateral occipital area as an additional, negative control site, because TMS to this region evokes muscle contractions indistinguishable from VWFA stimulation. rTMS was applied while subjects performed a lexical decision task on words and

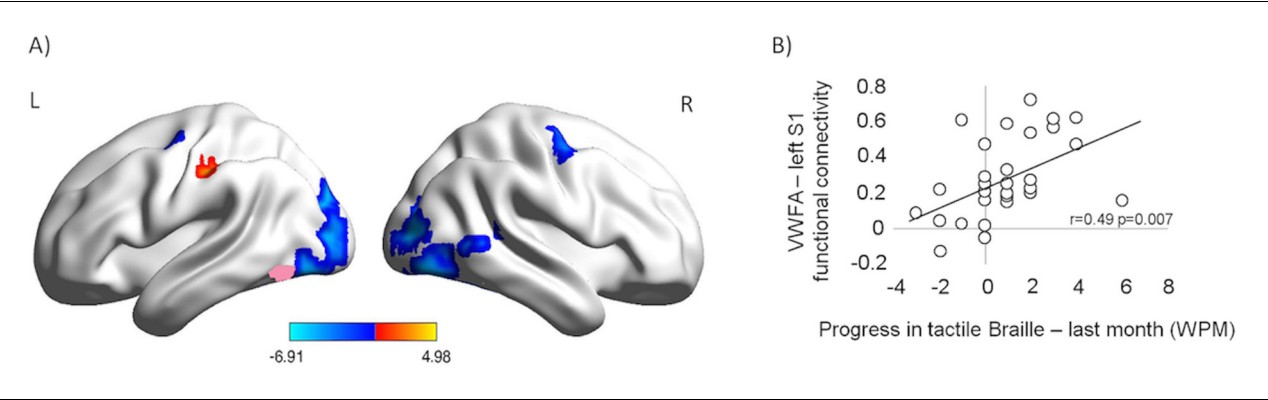

**Figure 3.** Following the tactile Braille course, the VWFA increased its resting-state connectivity with the somatosensory cortex while decreasing its coupling with other visual areas and the motor cortex. The connectivity between the VWFA and the somatosensory cortex was behaviorally relevant for tactile Braille reading. (A) Functional connectivity of the VWFA after the tactile Braille course relative to the before-course scan. Red represents increased correlation, and blue represents decreased correlation. The VWFA seed is marked in pink. Thresholds: p = 0.001 voxel-wise, p = 0.05 cluster-wise. (B) Correlation between after-course VWFA – left S1 functional connectivity and progress in tactile Braille reading speed in the last month of the course.

pseudowords written in tactile Braille (see *Figure 4A*; 'Materials and methods'). Based on a previous visual reading study (*Duncan et al., 2010*), we expected that stimulation to the VWFA during the performance of the task would decrease accuracy for lexical decisions on Braille words. As predicted, TMS to the VWFA decreased the accuracy of reading Braille words (t(8)=3.02, p=0.016, *Figure 4B*). This TMS result shows that the VWFA is necessary for reading tactile Braille words.

We also tested for effects of TMS on control stimuli (Braille pseudowords) and control sites (lateral occipital area, vertex) (*Figure 4B*) and none of these tests resulted in significant effects (VWFA, Braille pseudowords: t(8)=0.03, p=0.977; lateral occipital area, Braille words: t(8)=0.18, p=0.859; lateral occipital area, Braille pseudowords: t(8)=0.03, p=0.979; vertex, Braille words: t(8)=1.26, p=0.243; vertex, Braille pseudowords: t(8)=0.02, p=0.986). However, given the small number of subjects in this part of our study, our TMS experiment was underpowered to statistically verify the specificity of the TMS effect: ANOVAs testing for specificity of our effect of interest showed a trend for an interaction between the stimulus type and TMS for the VWFA (F(1,8)=3.59, p=0.095), no interaction between the stimulus type, TMS, and the stimulation site (F(2,16)=0.41, p=0.580), and no interaction between TMS and the stimulation site for Braille words (F(2,16)=1.5, p=0.253).

## Discussion

Several experiments have already indicated that in some contexts, the sighted's ventral visual cortex can contribute to the perception of tactile (reviewed in: *Amedi et al., 2005*) or auditory (*Amedi et al., 2007*) stimuli. It is also known that regions higher up in the sensory processing hierarchy, notably the antero-medial parts of the ventral temporal cortex (MNI y>-40), can host multisensory, abstract object representations (e.g. *Fairhall and Caramazza, 2013*; *Kassuba et al., 2014*). The left fusiform gyrus in particular is suggested to process object-specific crossmodal interactions (*Kassuba et al., 2011*). Our results demonstrate that the occipitotemporal visual cortex (VWFA, MNI y ≈ -60) can represent stimuli in a modality other than vision. The fact that TMS to the VWFA can disrupt tactile reading demonstrates the importance of this representation for sensory processing.

ROI analysis (*Figure 2B*) revealed that lateral occipital area (LOA) presented a pattern of activity increase to tactile words due to the course similar to the VWFA. However, TMS applied to LOA did not disturb tactile reading process (*Figure 4*). While the LOA is activated in various visual word recognition tasks (*Duncan et al., 2009*; *Wright et al., 2008*), its lesions seems not to affect reading itself (*Cavina-Pratesi et al., 2015*; *Milner et al., 1991*; *Philipose et al., 2007*). The increase of activity in both LOA and VWFA for tactile reading thus suggest that visual and tactile reading share similar neural correlates along the ventral visual stream. However, the exact function of LOA in reading seems to be more accessory than critical.

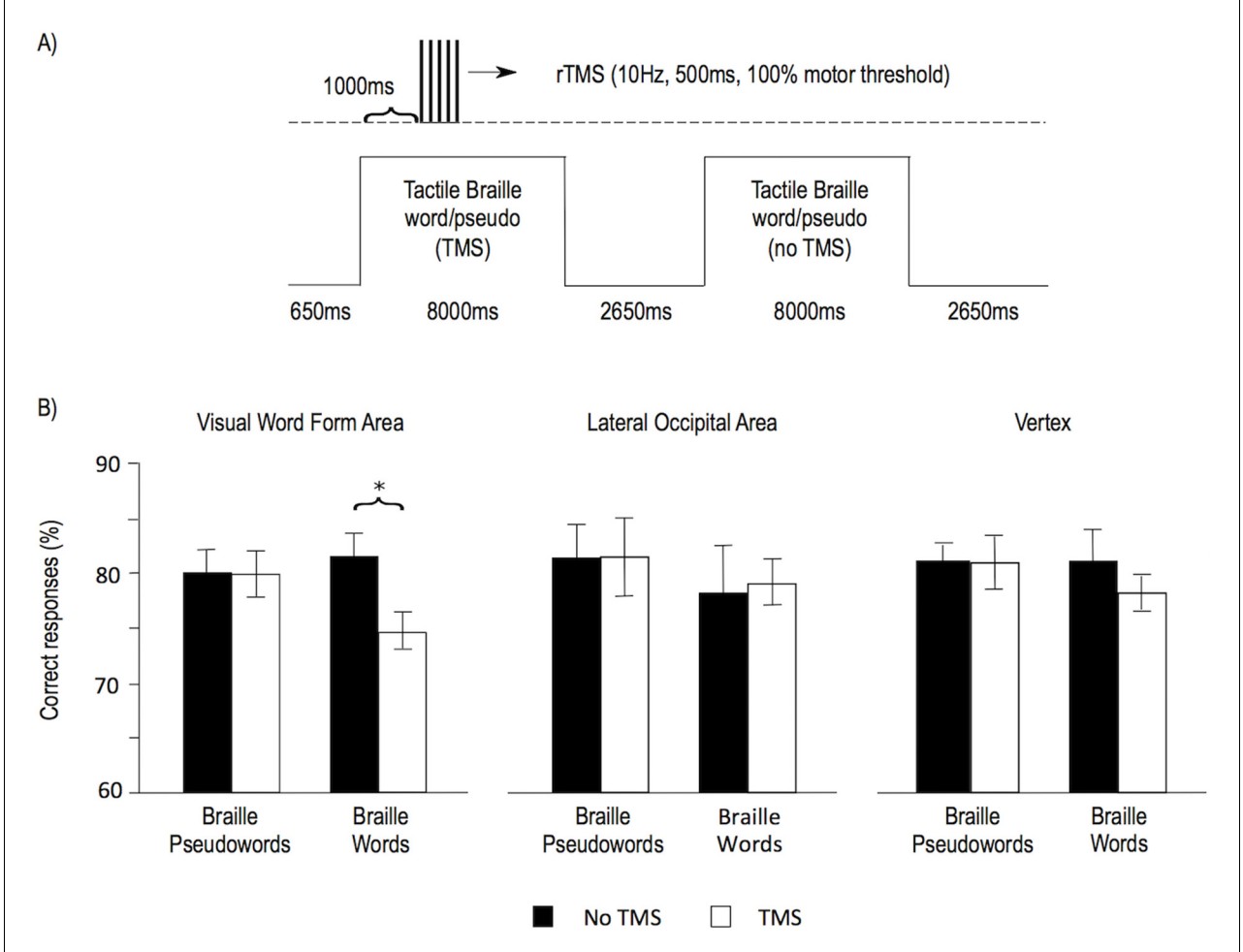

**Figure 4.** TMS applied to the Visual Word Form Area selectively decreased the accuracy of Braille word reading. (A) Illustration of the experimental design. Subjects read tactile Braille words or pseudowords and performed a lexical decision task based on them. In half of the trials, repetitive TMS was applied. The VWFA and two control sites (lateral occipital area and vertex) were tested in separate runs. (B) Mean accuracy of reading Braille words and pseudowords is shown for the VWFA and for both control sites, for the TMS and no TMS conditions separately. (*) p=0.016.

The ventral visual cortex was also activated when subjects heard auditory words and then imagined reading them in tactile Braille (*Figure 1—figure supplement 2, Table 4*). We also observed robust object activations in the object imagery task (ventral visual stream, left: MNI -45 -67 -9, Z=6.68, right: MNI 45 -67 -5, Z=5.94) which confirms that subjects successfully engaged in imagery during the experiment.

There are several reasons to rule out the possibility that the visual activation for tactile reading is a side-effect of mental imagery. First, if the visual activations were only a by-product of mental imagery, TMS to the VWFA would not interfere with Braille reading; yet, it did (*Figure 4B*). Second, training did not produce any changes in the activations to imagining tactile Braille (Appendix 1.5). Third, activations to imagining Braille in the VWFA did not correlate with tactile Braille reading proficiency (Appendix 1.4). Thus, visual cortex activations for tactile reading cannot be explained as a by-product of visual imagery. Instead, they constitute the signature of a new, tactile script representation emerging in the visual cortex.

Cortical reorganization that crosses the sensory boundaries is prominent in blind and deaf humans (*Hirsch et al., 2015*; *Pavani and Roder, 2012*; *Sadato et al., 2002*) and in congenitally deaf cats (*Lomber et al., 2011*). In the latter, the deprived auditory areas play a vital role in the deaf cats' superior visual perception. The above-mentioned studies used the same paradigms in non-deprived humans and cats, but did not find any signs of cross-modal reorganization (e.g. *Figure 7* in

**Table 4.** Summary of main activations in the control experiment.

| Contrast | Voxel-wise p treshold | Region | BA | Hemisphere | Z score | Cluster size | MNI coordinates | | |
|---|---|---|---|---|---|---|---|---|---|
| Tactile Braille Imagine vs Rest after course | p = 0.001 | Cerebellum | * | Right | >8 | 519 | 27 | -67 | -24 |
| | | | * | Left | 3.43 | 152 | -42 | -64 | -28 |
| | | Inferior Frontal Gyrus | 9 | Left | >8 | 8307 | -57 | 8 | 27 |
| | | Inferior Parietal Lobule | 40 | Left | >8 | | -39 | -43 | 43 |
| | | Medial Frontal Gyrus | 6 | Left | >8 | | -3 | -1 | 63 |
| | | Middle Frontal Gyrus | 46 | Left | >8 | | -42 | 32 | 23 |
| | | Postcentral Gyrus | 3 | Left | >8 | | -57 | -19 | 23 |
| | | Precentral Gyrus | 6 | Left | >8 | | -39 | -7 | 59 |
| | | Superior Temporal Gyrus | 22 | Left | >8 | | -54 | 8 | -1 |
| | | | 40 | Right | >8 | | 39 | -40 | 47 |
| | | Middle Occipital Gyrus | 19 | Left | 6.60 | 152 | -54 | -61 | -12 |
| | | Fusiform Gyrus (VWFA) | 37 | Left | 5.95 | | -45 | -61 | -12 |
| | | Inferior Temporal Gyrus | 20 | Left | 4.22 | | -51 | -49 | -16 |
| Visual Braille Imagine vs Rest after course | p = 0.001 | Inferior Parietal Lobule | 40 | Left | >8 | 3062 | -39 | -43 | 43 |
| | | | 40 | Right | >8 | 1233 | 39 | -40 | 47 |
| | | Middle Frontal Gyrus | 6 | Left | >8 | 3062 | -24 | -4 | 51 |
| | | | 6 | Right | 7.56 | 1233 | 27 | -4 | 51 |
| | | Precentral Gyrus | 6 | Left | >8 | 3062 | -54 | 2 | 39 |
| | | Cerebellum | * | Right | 7.63 | 134 | 30 | -67 | -24 |
| | | Inferior Frontal Gyrus | 44 | Right | 7.41 | 1233 | 54 | 8 | 23 |
| | | Middle Occipital Gyrus | 19 | Left | 7.01 | 119 | -48 | -58 | -12 |
| | | Fusiform Gyrus (VWFA) | 37 | Left | 6.38 | | -45 | -58 | -12 |
| | | Superior Temporal Gyrus | 42 | Right | 4.80 | 113 | 66 | -25 | 7 |
| Tactile Braille Imagine vs Visual braille imagine after course | p = 0.005 | Inferior Parietal Lobule | 40 | Left | 6.84 | 6602 | -57 | -22 | 23 |
| | | | 40 | Right | 4.72 | | 66 | -37 | 31 |
| | | Cerebellum | * | Right | 6.75 | 962 | 24 | -52 | -24 |
| | | Superior Frontal Gyrus | 10 | Left | 6.41 | 672 | -30 | 50 | 19 |
| | | | 10 | Right | 5.35 | 6602 | 27 | 44 | 19 |
| | | Precentral Gyrus | 4 | Left | 6.21 | | -33 | -19 | 55 |
| | | Postcentral Gyrus | 3 | Right | 5.72 | | 60 | -19 | 23 |
| | | | 2 | Left | 4.88 | | -57 | -25 | 47 |
| | | Middle Frontal Gyrus | 10 | Right | 5.21 | | 33 | 50 | 15 |
| | | | 9 | Left | 4.86 | 672 | -42 | 26 | 35 |
| Objects Touch | p = 0.005 | Cerebellum | * | Right | >8 | 1211 | 24 | -52 | -24 |
| | | | * | Left | >8 | | -21 | -55 | -24 |
| | | Postcentral Gyrus | 3 | Right | >8 | 5412 | 48 | -22 | 51 |
| | | Precentral Gyrus | 6 | Right | >8 | | 36 | -13 | 59 |
| | | | 4 | Left | >8 | | -39 | -22 | 59 |
| | | Middle Occipital Gyrus | 19 | Left | 5.18 | 85 | -51 | -64 | -9 |
| | | | 37 | Right | 4.77 | 1211 | 51 | -64 | 12 |

*Sadato et al., 2002*). One could have thus expected that learning to read by touch would lead to the emergence of a tactile reading area in the somatosensory cortex. Yet, our study produced a different result. Specific responses to tactile reading emerged not in somatosensory areas, but in visual areas.

This result might seem incompatible with a large body of data showing that tactile training leads to changes in somatosensory activation to tactile stimuli (*Guic et al., 2008*; *Kupers et al., 2006*; *Pleger et al., 2003*; *Ptito et al., 2005*), or decision-level frontal cortex changes (*Sathian et al., 2013*), but no visual cortex changes. However, all the above-mentioned experiments used simple stimuli, such as gratings, and short learning periods. Our experiment was longer (9 months) and used complex stimuli - entire Braille words. Experiments that study learning-related plasticity at multiple time points (*Lövdén et al., 2013*) suggest that at the initial stage of Braille learning, the somatosensory cortex might have increased its response to Braille words. Then, as the effects of early sensory learning consolidated in the somatosensory cortex, the cortical focus of learning shifted elsewhere, in our case the ventral visual stream.

Previous studies (*Büchel et al., 1998*; *Lomber et al., 2011*; *Merabet and Pascual-Leone, 2010*; *Sadato et al., 2002*, *1996*) have suggested that cross-modal plasticity is possible mainly as a result of massive sensory deprivation or injury. Our results demonstrate that given a long training period and a complex task, the normal brain is capable of large-scale plasticity that overcomes the division into separate sensory cortices. The exact mechanisms of this plasticity might of course be different in deprived and non-deprived subjects.

Earlier reports of cross-modal plasticity in the normal brain might have already hinted at the possibility of cross-modal plasticity in the normal brain. Indeed, it was shown that some parts of the visual cortex can be activated during auditory or tactile tasks (*Amedi et al., 2007*; *Kim and Zatorre, 2011*; *Powers et al., 2012*; *Saito et al., 2006*; *Zangenehpour and Zatorre, 2010*). Professional pianists, for example, were shown to activate their auditory cortex when viewing mute video recordings of a piano key being pressed (*Haslinger et al., 2005*). The behavioral relevance of such activations, however, was demonstrated only for simple tactile stimuli: grating orientation (*Zangaladze et al., 1999*) and distance judgment (*Merabet et al., 2004*). Our study used a controlled, within-subject design and precise behavioral measures supplemented with a causal method, TMS. Its results suggest that large-scale plasticity is a viable mechanism recruited when learning complex skills. This conclusion is congruent with a recent comparative study showing that the morphology of cerebral cortex is substantially less genetically heritable in humans than in chimpanzees (*Gomez-Robles et al., 2015*). Such relaxed genetic control might be the reason for homo sapiens' increased learning abilities and greater behavioral flexibility.

Despite evidence for cross-modal plasticity of the human brain, the dominant view still describes it as necessarily constrained by the sensory boundaries (e.g. Figure 18–2 in: Kandel et al., 2012). Our study provides a clear-cut evidence that learning-related neuroplasticity can overcome the sensory division. This calls for a re-assessment of our view of the functional organization of the brain.

## Materials and methods

### Subjects

Thirty-six subjects took part in the first fMRI study (32 females, 4 males; mean age = 29.17). All were right-handed, fluent in Polish and had normal or corrected-to-normal vision.

They were either Braille teachers/professionals (14 subjects), special education students specializing in blindness and related disabilities (11 subjects), or close relatives of blind people (4 subjects). All subjects had or was close to obtain higher education. All subjects showed high motivation to pursue the course. All but three subjects were familiar with Braille visually (see Appendix 1.2). However, all were naive in tactile Braille reading, which was verified by the baseline testing session (see Appendix 1.1). To ensure appropriate statistical power, prior to data collection we decided to recruit at least 30 subjects. The research described in this article was approved by the Commitee for Research Ethics of the Institute of Psychology of the Jagiellonian University (decisions 28/06/2012 and 12/03/2014). Informed consent and consent to publish were obtained from each of the participants in accord with best-practice guidelines for MRI and TMS research.

During the first fMRI examination, one subject was eliminated from the study due to a medical condition discovered by the neurologist supervising the study. Right after the first session, two subjects resigned for personal reasons. Another four subjects resigned during the tactile Braille reading course. Thus, 29 subjects completed the tactile Braille course and were included in the data analysis.

At the end of the course, nine of the subjects mentioned above who achieved tactile reading speeds between 3 to 17 WPM participated in the TMS experiment. All of them were female, and their ages ranged from 22 to 36 (M = 25.78, SD = 4.44).

## Tactile Braille course and behavioral measurements

For the purpose of this experiment, we developed a made-to-measure tactile Braille course. A detailed description of the course will be published elsewhere. Briefly, the course was based on individual exercises, to be performed while blindfolded. Subjects were instructed to train on one A4 sheet every day (approximately 20 min). The subjects' progress was monitored by qualified Braille teachers during monthly meetings. The course lasted 9 months. Teaching began with simple tactile recognition exercises. Next, Braille letters were gradually introduced. In the second half of the course, subjects were encouraged to focus mainly on whole-word reading. Although our Braille course was focused on tactile reading, all the subjects checked their exercises visually, which required visual Braille reading training on a daily basis. Thus, we assumed that they would also progress in visual Braille reading.

### Tactile Braille reading test

The subjects' tactile reading speed was tested monthly, starting from the 5th month of the course. The test consisted of 40 3–6 letter words (frequency of occurrence higher than 1 per million, according to SUBTLEX-PL; *Mandera et al., 2014*), to be read aloud blindfolded. The task was to read aloud as many words as possible in 60 s. The reading strategy was not specified. If a word was read inaccurately, it was not counted in the final score. Five different versions of the test were used during the course, each containing different words. Because we expected that word length would have a significant impact on reading speed (*Veispak et al., 2013*, *2012*), the word lists were ordered such that in each version of the test, words of a specific length appeared in exactly the same order (e.g. a three-letter word followed by a five-letter word, and so on). The distribution of word frequency was also similar across all versions of the test. Additionally, during each testing session, we measured a number of single Braille letters identified in one minute. The testing procedure and timing was identical to those in the whole-words reading test. The task was to read aloud as many letters as possible in 60 s. The order of the letters was counterbalanced between different versions of the test.

### Visual Braille reading test

The speed and accuracy of visual Braille reading were measured with a lexical decision task. The subjects were instructed to visually read Braille strings appearing on the screen and decide, as fast and as accurately as possible, whether they formed a valid Polish word or not and to indicate their choice with a mouse button. Subjects completed the lexical decision task at the beginning and at the end (i.e. in the 9th month) of the course.

The complete set of stimuli consisted of 320 items (160 Polish words and 160 pseudowords). All words were of low-to-moderate frequency (from 1 to 10 occurrences per million). The neighborhood size, as defined by the OLD20 measure (*Yarkoni et al., 2008*) was equated between words and pseudowords. We used SUBTLEX-PL to obtain the psycholinguistic characteristics of the items (*Mandera et al., 2014*).

Each trial began with a fixation cross displayed at the center of the screen for 500 ms, followed by a white screen for 150 ms. Subsequently, one item was presented (word or pseudoword), which lasted on the screen for 8000 ms or until a response was given. Afterwards, a white screen was displayed for 1000 ms, and the next trial started. Items were presented in a random order. Responses were provided via pressing the left or the right button of the computer mouse. Target buttons for words and pseudowords were counterbalanced between participants.

## fMRI data acquisition

All fMRI data were acquired on the same Siemens MAGNETOM Tim Trio 3T scanner (Siemens, München, Germany). The data from the fMRI reading experiment were collected using a 12-channel head coil. Resting-state fMRI data were acquired with a 32-channel head coil.

All data were collected using the same scanning parameters. We used a gradient-echo planar imaging sequence sensitive to blood oxygen level-dependent (BOLD) contrast (33 contiguous axial slices, phase encoding direction=posterior-anterior, 3.6 mm thickness, TR=2190 ms, angle=90°, TE=30 ms, base resolution=64, phase resolution=200, matrix=64x64, no iPAT).

The fMRI reading experiment was divided into four runs: the first two composed the main reading experiment (282 functional images for each run); the latter two composed the control imagery experiment (346 images in the first run, and 353 images in the second run). The resting-state data were collected in a separate scanning session, in a single run (282 volumes). In each scanning session, T1-weighted images were also acquired for anatomical localization.

To make acquisition conditions as similar as possible between the before- and after-training scans, in the before-course scan we measured each subject's head position relative to the head coil. Then, in the after-course scan, we reproduced the previously measured position. In addition, we used a standard Siemens Automatic Alignment scout MRI sequence before the two scans.

## fMRI stimuli and procedures

The fMRI reading experiment consisted of two parts: the main experiment (*Figure 1*) and the control mental imagery experiment (see *Figure 1—figure supplement 1*). In both experiments, we used a custom-made fiberglass table for the presentation of tactile stimuli. The table was designed in a way that prevented the subjects from seeing the stimuli that were placed on it. To minimize the time needed for the presentation of tactile stimuli, we used a block design in both parts of the reading experiment. Stimulation was programmed in Presentation (Neurobehavioral Systems, San Francisco, CA).

In both experiments, the blocks were separated by 13–20 s rest periods. Each rest period started with an ascending sound, which signaled to the subjects that they should raise their fingers from the fiberglass table. Then, the experimenter switched the cardboard with the stimuli. To equalize all experimental conditions, an empty cardboard was slipped onto the table for visual trials as well. At the end of the rest period, subjects heard a 500 ms descending sound signifying that they should put their fingers back down on the table. (In the imagery experiment, this sound was preceded by the auditory cue, e.g. 'imagine objects'.) To prevent them from touching the tactile stimuli prematurely, the subjects were asked to refrain from touching the cardboard until the metronome sound (main experiment) or the auditory cue (control experiment) was heard, 4–7 s later.

### Main experiment

In the main experiment, subjects were presented with Polish words and control stimuli. We used seventy-two 3- to 6-letter-long Polish words with a frequency higher than 1 occurrence per million according to SUBTLEX-PL; *Mandera et al., 2014*). All had low imageability scores (100–400 according to the MRC database; *Wilson, 1988*). There were six experimental conditions, each repeated four times. The conditions were: visually displayed Braille words (VB), visually displayed regular visual words (VW), words in tactile Braille (TB) and suitable control conditions. The words were counterbalanced between the conditions. The control conditions were: strings of 4–6 meaningless Braille signs composed of all 6 Braille dots presented either tactually or visually and strings of 4–6 hash signs (#) presented visually. TB words and tactile control stimuli were presented on cardboard sheets slipped onto the fiberglass table by the experimenter.

Each block contained 8 stimuli. Each stimulus was presented for 3500 ms, followed by a 500 ms fixation dot. The subjects heard a metronome sound at the onset of every stimulus. For the tactile conditions, the sound indicated that the subject should move his/her fingers to the next stimulus. We introduced this pace-setting manipulation because for tactile conditions, all 8 stimuli in a given block were presented on a single cardboard sheet. This pace-setting manipulation ensured that all eight stimuli were processed. Total block duration was 32 s.

## Control imagery experiment

The control imagery experiment was run immediately after the main experiment. In this experiment, the subjects first heard auditory words and then had to either read/touch or imagine them in TB, VB, and VW (*Figure 1—figure supplement 1A,B,D*). Three of the conditions: TB, VB, VW included the same stimulus types as the main experiment. The fourth condition (tactile objects) required the participants to touch objects: a plastic knife, a toothbrush, a plastic laundry clip, and a paintbrush, which were attached to a cardboard sheet and presented in a manner similar to the other tactile stimuli. Each of the four conditions was presented in two tasks (*Figure 1—figure supplement 1A*). In the first one, the tactile/visual stimulus was present, and the subject had to either read or touch it. In the second one, no tactile/visual stimulus was present, and the subject had to imagine it (*Figure 1—figure supplement 1D*).

Each block started with an auditory instruction indicating the task of the following block ('read', 'touch' or 'imagine') and the condition to be presented or imagined (VW, VB, TB, or objects; *Figure 1—figure supplement 1B*). There were 6 stimuli per block, except for the object conditions, which included a single stimulus only. Each stimulus was preceded by a 500 ms fixation dot and by its auditory description (e.g. 'toothbrush'). In the imagery conditions, this description was the only stimulus presented (the fixation cross remained on the screen). In the reading/touching condition, the auditory description was followed either by VW or VB words presented visually or by TB words or objects presented tactually. The stimulus presentation/imagining time was 3500 ms for words and 24,000 ms (the entire block) for objects.

## Resting-state fMRI

A 10-min long resting-state scan was performed according to standard protocols. During the scan, the subjects were asked to fix their gaze on the point displayed on the screen and relax but to refrain from sleeping.

## fMRI data analysis

The data from the fMRI reading experiment were analyzed using SPM8 software (www.fil.ion.ucl.ac.uk/spm/software/spm8/). Data preprocessing included: 1) slice timing, 2) realignment of all EPI images from the before-course and the after-course scans together, 3) coregistration of the anatomical image from the first time point to the mean EPI image, 4) segmentation of the coregistered anatomical image, 5) normalization of all images to MNI space and 5) FWHM spatial smoothing (5 mm). The signal time course for each subject was modelled within a general linear model (*Friston et al., 1995*) derived by convolving a canonical hemodynamic response function with the time series of the experimental stimulus categories and estimated movement parameters as regressors. Statistical parametric maps of the t statistic resulting from linear contrasts of each stimulus type minus baseline were generated and stored as separate images for each subject. Contrast images were then entered into an ANOVA model for random group effect analysis. We used first level contrast of each reading condition vs. its respective control after vs. before the tactile Braille course to assess the interaction between the time point (before and after the course) and reading condition (*Figure 1F*). The SPM8 paired t-test was applied in pairwise comparisons of the reading conditions and their respective controls (e.g. tactile Braille vs. tactile Control) and in pairwise comparisons across two time points (before and after the Braille course).

In addition to the GLM-based activation analysis, we used SPM8 regression to examine 1) how tactile Braille reading proficiency modulated activations during tactile Braille reading (*Figure 1G, Figure 1—figure supplement 3D*) and during tactile and visual Braille imagining and 2) how visual Braille reading speed influenced activations in tactile Braille reading.

We applied a voxel-wise threshold of $p<0.005$ and a $p<0.05$ threshold for cluster extent, corrected for multiple comparisons using REST AlphaSim (1000 Monte Carlo simulations), across the whole brain, unless stated otherwise. Similar results were observed at a voxel-wise threshold of $p<0.001$, though not necessarily at cluster level-corrected levels of significance.

## Representation similarity analysis

We applied representation similarity analysis (RSA) (*Kriegeskorte et al., 2008*) to compare the neural code of the three tested conditions: visual words, visual Braille, and tactile Braille. The individual

visual word form area (VWFA) functional ROIs were defined as a set of 100 voxels that presented the highest t-value in the contrast of visual Word reading vs. visual word control before the course within the broad limits of the possible VWFA, that is, within a box of coordinates with the range: -50>x>-25, -68>y>-40, -25>z>-5. Inside this functional ROI, condition-related activity patterns were extracted for the three conditions of interest using the corresponding unsmoothed contrast maps: visual reading (visual words vs. control), visual Braille reading (visual Braille vs. control) and tactile Braille reading (tactile Braille vs. control), all after the course. Thus, for each voxel in the ROI, we extracted beta values for each reading condition. Then, those beta values were correlated pairwise (visual words x visual Braille, visual Braille x tactile Braille and tactile Braille x visual Braille). As a result, for each participant, we obtained a neural similarity matrix describing how representationally similar the activity patterns associated with the three conditions were. The resulting correlation coefficients for each condition pair were Fisher *r-*to-*z* transformed (z(r)) and compared on a group level in one-way repeated measures ANOVA with a factor of script pairs (3 levels: tactile Braille and visual Braille, tactile Braille and visual words, and visual Braille and visual words). Simple effects were analyzed using post hoc tests with Bonferroni correction. For simplicity of interpretation, in the main text we report the correlation coefficients (*Andrews-Hanna et al., 2007*).

To avoid double-dipping (*Kriegeskorte et al., 2009*), the ROIs were extracted from the contrast of visual word reading before the course, whereas the contrasts used for RSA were all taken after the course. Results similar to those presented in *Figure 2A* were obtained from the contrast of all reading conditions vs. their controls and from purely anatomically defined ROIs.

## ROI analysis

To avoid double-dipping (*Kriegeskorte et al., 2009*), in the ROI analysis of the main reading experiment, all ROIs were defined (*Figure 2B–E*) based on contrasts from the separate imagery experiment (*Figure 1—figure supplement 1A*).

ROIs were defined as 3x3 voxel cubes positioned at the peak of the relevant imagery experiment contrast, masked by an anatomical mask for the region in question (see below). The resulting peak MNI coordinates for *Figs 2B–E* are shown in the captions of each ROI subplot. For *Figs 2B and 2E*, we used the visual word reading contrast. In the case of *Figure 2B*, the y coordinate of the ROI was tethered at y=-57, the canonical y coordinate of the VWFA (*Cohen et al., 2002*), to avoid a bias toward non-specific visual activation. Such a bias would have otherwise led to the selection of a much more posterior ROI (the imagery experiment did not contain visual control stimuli that could have been used to correct this bias). For *Figs 2C–D* and for the secondary somatosensory (SII) and primary motor (MI) cortex results reported in the text, we used the object touch contrast minus the resting baseline. For *Figure 2C* (Lateral Occipital tactile visual area; *Amedi et al., 2001*), this contrast was further constrained by an anatomical mask of BA37. For the primary somatosensory cortex (*Figure 2D*), the activation was additionally constrained by a 15 mm sphere centered on coordinates reported in the literature as corresponding to the part of the primary somatosensory cortex that hosts the finger representation modified during tactile training (MNI -54, -20, 48; *Pleger et al., 2003*; similar ROI results were obtained for other SI definitions, such as constraining by a simple anatomical mask of primary somatosensory cortex). For the secondary somatosensory cortex (see main text), we used a mask made by merging Brodmann Areas 40 and 43 (WFU PickAtlas, http://www.fil.ion.ucl.ac.uk/spm/ext/) and further constraining them to the ceiling of the lateral sulcus, where the secondary somatosensory cortex is located (parietal operculum – see *Eickhoff et al., 2006*; *Ruben et al., 2001*). The error bars in *Figure 2B–E* represent the SEM across subjects after subtraction of the individual subjects' mean.

## Resting-state fMRI

Data Processing Assistant for Resting-State fMRI (DPARSF; *Chao-Gan and Yu-Feng, 2010*) and SPM8 (www.fil.ion.ucl.ac.uk/spm/software/spm8/) were used to process the data. The first 10 volumes of each subject's scan were discarded for signal stabilization and for subjects' adaptation to scanner noise. Then, slice-timing correction and head-motion correction were applied. The magnitude of participant head motion was quantified by computing mean relative displacement (*Van Dijk et al., 2012*) and mean frame displacement (*Power et al., 2012*) measures. Both measures showed no difference in the magnitude of head motion between the first and second scanning session. T1

images were segmented, and both anatomical and functional images were normalized to MNI space.

Two steps specific to the functional connectivity analysis - regression of nuisance covariates and bandpass filtering - were performed to reduce spurious variance unlikely to reflect neuronal activity. Nuisance regression was performed first to avoid attenuation of the signal caused by mismatch in the frequencies of the data and regressors (*Hallquist et al., 2013*). Nuisance variables included: a) white matter signal; b) cerebrospinal fluid signal; c) 24 head motion parameters: 6 parameters of the current volume, 6 of the preceding volume and a quadratic term for each of these values (*Friston et al., 1996*, see also: *Satterthwaite et al., 2013*; *Yan et al., 2013*); and d) a separate regressor for every volume that displayed a mean frame displacement value higher than 0.5 (*Power et al., 2012*). Given recent evidence that global signal regression can disturb meaningful signal (*Fox et al., 2009*; *Murphy et al., 2009*; *Weissenbacher et al., 2009*), we did not include global signal as a regressor. After the regression of nuisance covariates, a bandpass filter (0.01–0.08 Hz) was applied. The resulting images were smoothed with a 5 mm FWHM Gaussian kernel.

For the whole-brain functional connectivity analysis, we defined a VWFA seed of interest (*Figure 3*) based on the fMRI activations shown in *Figures 1–2*. The seed was defined as the 20 most active voxels in [(tactile Braille vs. Control) x (after training > before training)] contrast (*Figure 1E*), in the left fusiform gyrus and inferior temporal gyrus regions (the mask was created using Harvard-Oxford Cortical Structures Atlas). The somatosensory cortex seed was defined as a sphere with a 4 mm radius that was centered on the same ROI as used for the ROI analysis shown in *Figure 2D*. Similar results were obtained with different ROI sizes (e.g. spheres with a 6 mm or 8 mm radius).

The functional connectivity measure (FC) was calculated according to standard procedures. In the whole-brain analysis, for each seed, subject, and scan, we computed a voxel-wise correlation of time courses between seed regions and the rest of the brain, which was then transformed into Fisher's z value maps. In the ROI analysis, a BOLD time course was extracted for each seed, subject, and scan. The seed time courses were then correlated, creating a Pearson's r correlation coefficient for each subject and scan. Fisher's z transform was applied, to ensure normality for the t-test. Correlation coefficients were then correlated with the behavioral measure. Previous studies show that training can lead to rapid change in resting-state functional connectivity pattern (e.g. *Lewis et al., 2009*; *Urner et al., 2013*; *Voss et al., 2012*). We thus expected that final resting-state functional connectivity reflects subjects' intensity of training in last days of the Braille course. This can be quantified as a change in Braille reading speed over last month of the course. This behavioral measure was thus used for correlation. Given our apriori hypothesis, uncorrected significance value is reported. However, the same result was obtained even when correction for multiple comparisons was applied to account for multiple behavioral sessions.

Whole-brain analysis was thresholded at p=0.001 voxel-wise and p=0.05 cluster-wise. The Brain-Net Viewer toolbox was used for data visualization (*Xia et al., 2013*).

## The TMS experiment

### Localization of TMS targets

Three brain targets were chosen for neuro-navigated TMS, namely the VWFA and two control sites—the vertex and the lateral occipital area (LO). We chose the LO as an additional control site for the VWFA stimulation because it is a visual recognition region in proximity to the VWFA, and its stimulation produces the same unspecific effects. At the same time, TMS studies have shown that the LO is not engaged in the recognition of visually presented words (*Duncan et al., 2010*).

The VWFA and the LO were marked on each subject's structural MRI scan based on individual fMRI data from reading experiments. The VWFA target was chosen as the before-course peak of the visual word reading versus the control condition in the main fMRI experiment (*Figure 1B*, *Figure 1—figure supplement 3A*;*Dehaene and Cohen, 2011*), restricted to the left fusiform gyrus. The LO target was chosen as the before-course peak of the tactile object recognition vs. baseline contrast in the control imagery fMRI experiment (*Figure 1—figure supplement 1A*; *Amedi et al., 2001*). The vertex was localized anatomically, on the T1 image of each subject.

## Task

A lexical decision task was used to measure the speed and accuracy of tactile Braille reading. Subjects were instructed to tactually read Braille strings appearing on a Braille display and to indicate whether they formed a valid Polish word as fast and as accurately as possible by pressing a key on the response pad (see 'procedure' below). Each trial began with a 650 ms blank period. Subsequently, one item was presented (a word or a pseudoword), which was displayed for 8000 ms or until the response was given. A 2000 ms blank period ensued, followed by the next trial. Items were presented in pseudorandom order. The task consisted of 360 trials, split into three runs. The runs lasted 10–15 min each, depending on the subject's tactile reading speed. Responses were provided by pressing the left or right button on the response pad. Target buttons for words and pseudowords were counterbalanced between participants. Task difficulty was optimized for accuracy analysis, and the variability in accuracy was high (62–92%). Thus, we did not expect to find significant results in the reaction time analysis, which was confirmed with statistical tests.

Stimulus presentation and response logging were programmed in Presentation (Neurobehavioral Systems). Stimuli were displayed using the BraillePen 12 Touch (Harpo, Poznań, Poland), integrated with Presentation via in-house Presentation code. Responses were provided using the RB-830 Response Pad (Cedrus, San Pedro, USA).

## Materials

The complete set of stimuli used in the TMS experiment consisted of 360 items (180 Polish words and 180 pseudowords). All items were three to six letters long. All words were of low-to-moderate frequency (from 1 to 20 occurrences per million, M = 5.08, SD = 5.06). The neighborhood size, as defined by the OLD20 measure (*Yarkoni et al., 2008*), was equivalent between words and pseudowords. Words and pseudowords were further divided into two lists, equivalent in length and neighborhood size; words were additionally matched for frequency. Items from one list were tested in the TMS condition (i.e. TMS was applied when they were being shown), whereas items from the other list were tested in the no-TMS control condition (i.e. no TMS was applied when they were being shown; *Figure 4A*). The lists were counterbalanced across subjects. The SUBTLEX-PL (*Mandera et al., 2014*) linguistic corpus was used to obtain the psycholinguistic characteristics of the items.

Pseudowords were constructed either by mixing letters from words used in the experiment (in words composed of one syllable) or by mixing syllables taken from words used in the experiment (in words composed of more than one syllable). We included only items that did not form a valid word but that were phonologically and orthographically plausible.

## TMS protocol

Repetitive TMS (rTMS) was applied pseudorandomly during half of trials in each run. During each run, TMS was delivered to one of three target sites – the VWFA, the LO or the vertex. Participants were not aware of the order of the target sites, and the order was counterbalanced across participants. TMS pulses were delivered at 1000, 1100, 1200, 1300, and 1400 ms after stimulus onset (i.e., 10 Hz for 500 ms). We decided to delay the TMS for 1 s because of the speed of participants' tactile reading – we assumed that 1 s was sufficient to complete the early motor phase and to start the processes specifically linked to reading, which we wanted to disrupt in the experiment. The intensity of TMS was set to 100% of the resting motor threshold, defined as the lowest intensity needed to elicit a visible twitch of the hand that was kept relaxed by the participant. The TMS protocol used in this experiment is in agreement with established safety limits (*Rossi et al., 2009*). Similar protocols have been widely used to interfere with processing in the VWFA (*Duncan et al., 2010*) as well as in other brain regions (*Göbel et al., 2001*; *Gough et al., 2005*; *Sandrini et al., 2008*).

A MagPro X100 stimulator (MagVenture, Hückelhoven, Germany) with a 70 mm figure-eight coil was used to apply the TMS. Moreover, a frameless neuronavigation system (Brainsight software, Rogue Research, Montreal, Canada) was used with a Polaris Vicra infrared camera (Northern Digital, Waterloo, Ontario, Canada) to guide stimulation.

## Procedure

Participants were asked to fill out safety questionnaires. Next, they were familiarized with TMS, and the resting motor threshold was measured using single pulses applied to the hand area of the left primary motor cortex. Afterwards, subjects were asked to put on a blindfold while they performed the training session without TMS to familiarize themselves with the task. This first training session was followed by the second one, when participants were performing the task with rTMS to get used to the stimulation. After both training sessions, one target site was chosen and the main experiment began. All three target sites were tested one-by-one, in three separate runs, with 5 min breaks between them. The whole TMS experiment lasted approximately 90 min. The items used in the training session (words/pseudowords) were not used in the main session.

## Acknowledgements

Supported by a National Science Centre Poland grant (2012/05/E/HS6/03538), a Marie Curie Career Integration grant (618347) and funds from the Polish Ministry of Science and Higher Education for co-financing of international projects, years 2013-2017, awarded to MS. AM and KJ were supported by a grant from the National Science Center Poland (2014/14/M/HS6/00918). This project was realized with the aid of CePT research infrastructure purchased with funds from the European Regional Development Fund as part of the Innovative Economy Operational Programme, 2007–2013. We gratefully acknowledge Boris Gutkin, Christophe Pallier, Antonio Moreno, Valentina Borghesani, Karim N'dyaie, Małgorzata Kossut, Weronika Dębowska, Adam Ryba, Karolina Dukała, Avital Hahamy, Paweł Hanczur, the subjects, and the Polish blind community.

## Additional information

### Funding

| Funder | Grant reference number | Author |
| --- | --- | --- |
| National Science Centre Poland | 2012/05/E/HS6/03538 | Marcin Szwed |
| Marie Curie Career Integration Grant | 618347 | Marcin Szwed |
| Ministerstwo Nauki i Szkolnictwa Wyższego | | Marcin Szwed |

The funders had no role in study design, data collection and interpretation, or the decision to submit the work for publication.

### Author contributions

KS-K, ŁB, MS, Conception and design, Acquisition of data, Analysis and interpretation of data, Drafting or revising the article; MP, ES, Designed and taught the Braille course; KJ, AM, MWŚ, Additional contributions to acquisition of data; AA, Additional contributions to conception and design, data analysis, and revising the article

### Author ORCIDs

Marcin Szwed, http://orcid.org/0000-0002-2153-7793

### Ethics

Human subjects: The research described in this article was approved by the Commitee for Research Ethics of the Institute of Psychology of the Jagiellonian University (decisions 28/06/2012 and 12/03/2014). Informed consent and consent to publish were obtained from each of the participants in accord with best-practice guidelines for MRI and TMS research.

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

# Appendix 1 – Supplementary results

## 1.1. Tactile Braille reading speed

*Braille word reading speed prior to the onset of the course.* To make sure that our subjects were naïve in tactile Braille prior to the onset of the course, subjects were tested with the tactile Braille reading test before the first fMRI session (Materials and methods). From the 29 subjects that were included in the data analysis, 26 subjects were unable to read even a single word during the 60 s allowed in the test. One subject managed to read one word and two subjects read two words.

*Single-letter recognition speed prior to the onset of the course.* Additionally, we tested single Braille letter recognition, in a manner similar to that used for word reading (Materials and methods). The median number of Braille letters read per minute was 0 (mean=2.28), with individual scores ranging from 0 to 7 letters per minute. These results confirm that at the beginning of the course, the great majority of our subjects were utterly unable to read Braille by touch. In fact, their capacity to recognize even single Braille letters was also very limited.

*Braille word reading speed after the course* The distribution of final reading speed was normal (Shapiro-Wilk (29)=0.94, p = 0.123), with the mean=6.21, SD=3.94, median=5, mode=4, range 0-17. There were 2 subjects who read 0 words per minute at the end of the braille course.

*Single-letter reading speed after the course.* After the course, the subjects reached a mean performance of 18.41 single tactile letters read per minute (SD = 4.81). However, there was a large degree of variability in performance, with results ranging from 10 to 29 letters per minute.

*Demographic background and tactile Braille reading speed.* We tested if progress in tactile Braille reading was modulated by the demographic background of the subjects. To this end, we divided the subjects into three groups – Braille teachers/professionals (14 subjects), special education students specializing in blindness and related disabilities (11 subjects) and close relatives of blind people (4 subjects) – and entered this division into a regression analysis with final tactile Braille word reading speed as the dependent variable. However, we did not find any significant effect in the analysis (all p>0.2). To check if this insignificance was due to the evidence for the null hypothesis, or the insensitivity of the data, we computed Bayes factor with BayesFactor package implemented in R (*Rouder and Morey, 2012*). The lmBF function in the BayesFactor package estimates linear models and returns the Bayes factor (BF) of the model relative to a null model that predicts the data by the intercept alone. BF values greater than 1 would indicate evidence for the alternative over a null hypothesis, values less than 1 the converse, and BF close to 1 would indicate that data is insensitive (*Dienes, 2014*). BF equalled 0.35, therefore we can assume that there is a strong evidence towards the null hypothesis. We thus conclude that demographic background cannot predict the progress in tactile reading in our participants.

## 1.2. Visual Braille reading speed

While all of our participants were naïve in tactile Braille, most of them knew how to read Braille visually. Moreover, the course curriculum included learning to recognize Braille letters by sight. As a result, subjects also improved their visual Braille reading speed.

We measured our subjects' visual Braille reading speed with a lexical decision task in visual Braille (Materials and methods). The test was administered at the beginning and at the end of the course. Only subjects whose accuracy in each session was higher than 60% (chance level was 50%) were included in the reaction time analysis. On this basis, 3 subjects were excluded from the reaction time analysis. Moreover, one subject was excluded from both the accuracy

and reaction time analysis due to missing data from the second testing session. Thus, the accuracy analysis was performed on 28 subjects and the reaction time analysis, on 25 subjects.

The mean accuracy (calculated as the mean of the individual subjects' medians) was 78.25% (SD = 13.91%) at the beginning and 88.14% (SD = 8.75%) at the end of the course. The mean reaction time (RT) was 4587 ms (SD = 1071 ms) at the beginning and 3799 ms (SD = 846 ms) at the end of the course.

The difference between testing sessions was significant, both in the case of accuracy (t(27) = 4.72, p < 0.001) and of reaction time (t(24) = 7.05, p < 0.001). This result indicates that subjects also progressed in visual Braille reading during the Braille course.

Additionally, we transformed the reaction times for visual Braille words into a word per minute (WPM) measure to compare the effect of visual and tactile Braille proficiency on brain activity (Appendix 1.4). For this reason, we calculated each subject's median RT (as it is more robust to outliers than the mean) in visual Braille reading, and then we divided 60 s by this median RT. This measure resulted in an estimated number of words that the subject could read in one minute. This measure was then implemented in an SPM regression model.

## 1.3. Main experiment – activations to visual Braille and visual words

Before the course, visual word reading resulted in a cluster of activity within the left ventral visual stream, covering the inferior temporal gyrus, fusiform gyrus and the middle occipital gyrus (*Figure 1—figure supplement 3A*, *Table 2*). Interestingly, we found much more robust activity during visual Braille reading. Apart from the activity in the bilateral ventral occipital stream, this condition elicited activation in the bilateral frontal and parietal areas, predominantly in the left hemisphere, including the inferior frontal and precentral gyri, precuneus and middle temporal gyri (*Figure 1—figure supplement 3B*, *Table 2*). The VWFA activity for visual Braille reading was stronger than for visual words, similar to previously reported responses to novel visual alphabets (*Vogel et al., 2014*; *Xue et al., 2006*) Accordingly, the aforementioned visual Braille lexical decision task (see section 1.2) revealed a very slow recognition speed for visual Braille reading (detailed results will be published elsewhere).

The pattern of activity for visual words remained intact across the tactile Braille reading course. We found no suprathreshold voxels in the contrast of visual word reading after versus before the course, even at an exploratory threshold of p=0.01. The same comparison for visual Braille (before<after the course) revealed an after-course increase in activation in the bilateral precuneus, middle temporal gyrus, parietal lobules and mesial frontal areas (*Figure 1—figure supplement 3C*, *Table 2*). All of those areas are nodes of the default mode network (*Raichle et al., 2001*) and showed profound de-activation for all of the tasks in our experiment. Their increased engagement in visual Braille reading could suggest increased cognitive control over the process.

## 1.4. fMRI-behavior correlations

We used the behavioral measurements of tactile and visual Braille reading speed to see how proficiency in Braille reading can modulate the fMRI signal when performing a given reading task. We performed the following correlations:

*Tactile Braille reading activations – tactile Braille reading speed.* The result is reported in the main text (*Figure 1G*, *Table 3*).

*Tactile Braille reading activations – TB single-letter recognition speed.* We checked for signal modulation in tactile Braille reading after the course using the performance in single

tactile letter reading. We found that tactile single-letter reading speed correlated positively with the activations localized only in the left ventral visual pathway, including the lingual gyrus, middle and inferior occipital gyri, and the fusiform gyrus (*Figure 1—figure supplement 3D*, *Table 3*). This result further confirms the engagement of the left ventral visual stream in tactile reading.

*Tactile Braille reading activations – visual Braille reading speed.* Visual Braille reading speed did not significantly modulate any tactile Braille reading activations in the visual system.

*Tactile Braille imagery activations – tactile Braille reading speed.* We found no correlations between tactile Braille proficiency and the signal when it was imagined, even at an exploratory threshold of p=0.01 voxel-wise.

*Visual Braille imagery activations – tactile Braille reading speed.* Tactile Braille proficiency did not modulate activity during visual Braille imagery even at an exploratory threshold of p=0.01 voxelwise.

## 1.5. Control imagery experiment – fMRI results

In addition to the main experiment, our subjects also completed a control experiment aiming at exploring the imagery processes accompanying the process of tactile reading (see Materials and methods, and *Figure 1—figure supplement 1A,B,D*). In this experiment, their task was to read or imagine visual words, visual Braille words and tactile Braille words and to touch or imagine every-day-use objects.

*Tactile and visual Braille imagery.* Imagining tactile Braille words vs. rest after the course elicited robust activation in the bilateral cerebellum, bilateral precentral and postcentral gyri (predominantly in the left hemisphere), the left inferior and middle frontal gyri, the bilateral superior temporal sulcus and finally the left fusiform gyrus (MNI -45 -61 -12, BA37, Z = 5.95, p<0.001, *Figure 1—figure supplement 2A*, *Table 4*). A very similar pattern of activation was found for visual Braille imagine vs. rest after the course (*Figure 1—figure supplement 2B*, *Table 4*). Here, the fusiform peak was localized slightly more anteriorly: MNI -45 -58 -12 (BA37, Z = 6.38, p<0.001).

We then compared the two imagery conditions. For the visual Braille imagery vs. tactile Braille imagery contrast, we found no suprathreshold voxels, even at an exploratory level of p=0.01 voxel-wise). However, relative to visual Braille imagery, tactile Braille imagery caused more robust activation of the motor and somatosensory areas, the bilateral inferior parietal lobules and the cerebellum (*Figure 1—figure supplement 2C*). This finding is in accordance with data showing that motor imagery activates the primary motor and somatosensory cortices (*Lotze and Halsband, 2006*; *Porro et al., 1996*).

*Visual words imagery.* Imagining visual words before and after the course elicited activations in the medial frontal gyrus, left precentral gyrus and Broca's area (left inferior frontal gyrus, BA44) as well as in the left inferior parietal lobule and the right cerebellum. When we lowered the threshold to p=0.005, uncorrected, we found also a cluster in the left occipital lobe, peaking in the fusiform gyrus (MNI -45 -58 -12, BA37, Z = 4.73).

*Object imagery.* Imagining objects before and after the course activated primary visual areas (BA18, BA19) bilaterally, the bilateral precentral and postcentral gyri and bilateral frontal areas, including the medial frontal gyrus and inferior frontal gyri. Temporal activations were also bilateral, with a peak in the right superior temporal gyrus.

*Object touch vs. rest.* In the contrast of touching objects vs. rest, we found bilateral lateral occipital activations, which peaked in the middle occipital gyrus (left: MNI -51 -64 -9, BA 19, Z=5.19; right: MNI 51 -64 12, BA37, Z=4.77, *Figure 1—figure supplement 2D*). This result confirms the existence of a visuo-haptic area called the lateral occipital tactile-visual (LOtv) area reported by Amedi and colleagues (*Amedi et al., 2001*). This area is known to be

activated in visual and tactile object recognition tasks. Recently, it has also been shown to respond to shape information conveyed via the auditory modality through sensory substitution devices (*Amedi et al., 2007*).

*Effects of the Braille course on imagery.* Finally, none of the responses to the three imagery conditions showed any changes following the tactile Braille course. Interactions of the tactile Braille imagery, visual Braille imagery, and visual Word imagery contrasts with the after- vs. before-course states showed no suprathreshold voxels (neither positive nor negative), even at an exploratory threshold of p=0.01.

## 1.6. Representation similarity analysis: ANOVA results

Statistical analysis was performed on Fisher z-transformed correlation coefficients (z(r)) – see Materials and Methods. The main effect of scripts' pairs was highly significant (F(2,56)=14.53, p<0.001). Simple effects of scripts' pairs were assessed using post-hoc tests with Bonferroni correction. The correlation between the two Braille conditions (mean z(r)=0.59, SD=0.08) was significantly higher than the correlation between tactile Braille and visual Words (mean z(r) =0.28, SD=0.09, p<0.001) and visual Braille and visual Words (mean z(r)=0.11, SD=0.07, p=0.020). The correlation between visual Braille and visual words was higher than the correlation between tactile Braille and visual Words, however this result was only marginally significant (p=0.063).

## 1.7. Supplementary RS-fMRI analysis

*Whole-brain functional connectivity analysis with the VWFA as a seed region.* The results are reported in the main text (*Figure 3*)

*VWFA – left S1 functional connectivity ROI-based analysis.* To confirm the increase of VWFA – left S1 functional connectivity that was observed in the whole-brain analysis (*Figure 3*), we performed an independent ROI analysis of functional connectivity between independently defined VWFAs and left S1 ROIs (see Materials and methods). Functional connectivity between these two regions increased following the Braille course, from r=0.20 to r=0.29 (t(28) = 2.3, p = 0.029; statistical comparison was performed on Fisher z-transformed correlation coefficients – see Materials and Methods).

