## [Decision Letter]

Thank you for submitting your work entitled "Massive cortical reorganization in sighted Braille readers" for peer review at *eLife*. Your submission has been favorably evaluated by Timothy Behrens (Senior editor) and three reviewers, one of whom, Heidi Johansen-Berg, is a member of our Board of Reviewing Editors, and another is Krish Sathian.

The reviewers have discussed the reviews with one another and the Reviewing editor has drafted this decision to help you prepare a revised submission.

Summary:

This is a novel and interesting study that provides clear and converging evidence on multimodal plasticity following Braille training in sighted individuals. The authors are to be especially commended for their incorporation of multiple control conditions, using interaction analyses and avoiding double-dipping.

Essential revisions:

1) For the ROI analysis (Figure 2) the interaction in the VWFA (and IPS and LOA) is driven as much by increased deactivation to control stimuli as by increased activation for tactile Braille words. How should this be interpreted? Is there a significant increase in positive activation from pre to post training for tactile Braille words (i.e. as a post-hoc test to determine what is driving the interaction)?

2) TMS study – although the lateral occipital area was chosen here as a control site, doesn't the FMRI data (Figure 2) suggest it shows a similar effect to the VWFA? How should this be reconciled?

3) In the Discussion, it should be explained why several experiments failed to find similar reorganization in sighted subjects, even after extensive training (Kupers et al., 2006; Ptito et al., 2005).

4) Figure 1 shows a correlation between post-training activity during tactile Braille reading and Braille reading speed. How many subjects had a reading speed of 0 WPM? Is the spread of reading speed suitable for a linear regression analysis?

5) For the correlation between resting connectivity strength and change in reading speed over the preceding month (Figure 3) – how many behavioural variables were tested for correlation here and should some correction be made for multiple comparisons? Why change over the past month rather than some other time interval? And why change in reading speed rather than simply reading speed (when the latter was considered for Figure 1 for example)?

6) In the subsection “Representation Similarity Analysis”: The claim that the correlations of Braille conditions with the visual word condition were weaker than those between the two Braille conditions should be substantiated by appropriate statistical tests. The tests reported in Section 1.6 of the Appendix do not seem to address this issue.

7) The authors used a rather rapid rate of rTMS (10 Hz). Although this has indeed been used to suppress activity as in the prior study of rTMS over the VWFA, this rate is sometimes considered facilitatory. Could the authors comment on this?

8) In the Introduction, considerable attention is devoted to the idea that the behavioral relevance of the cross-modal recruitment of visual cortex demonstrated in the present study is highly novel. Without detracting from the novelty of the study, it would be appropriate to acknowledge earlier studies that did demonstrate such behavioral relevance, e.g., Zangaladze et al., Nature, 401: 587-590, 1999; Merabet et al., Neuron, 42: 173-179, 2004.

---

## [Author Response]

Essential revisions: 1) For the ROI analysis (Figure 2B) the interaction in the VWFA (and IPS and LOA) is driven as much by increased deactivation to control stimuli as by increased activation for tactile Braille words. How should this be interpreted? Is there a significant increase in positive activation from pre to post training for tactile Braille words (i.e. as a post-hoc test to determine what is driving the interaction)?

Thank you for helping us to clarify this important issue. Our analysis, which we outline below, shows that the interaction in the VWFA and LOA is driven by two effects. First, there is a decrease of activation to tactile control stimuli. Second, there is an increased activation for tactile Braille words. Post-hoc tests show that none of the two effects are significant on their own. In the VWFA, where the interaction is significant at p<0.001, the post-hoc paired t-tests show p=0.23 for tactile Braille words (after course) vs. tactile Braille words (before course) and p=0.38 for tactile control (after course) vs. tactile control (before course). In the LOA, where the interaction is significant at p=0.0047, the post-hoc paired t-tests show p=0.35 for tactile Braille words (after course) vs. tactile Braille words (before course) and p=0.41 for tactile control (after course) vs. tactile control (before course). In both locations, the interaction is driven by both factors. Per your request, we have clarified the above in the new version of the manuscript, with the following change to the description of the ROI results in the VWFA and the LOA:

“In the VWFA (Figure 2; all ROIs are in the left hemisphere) following the Braille course, the response to tactile Braille words changed from de-activation to positive activation, resulting in a significant difference between tactile words and their control (interaction: p<0.001). […] These responses remained unchanged throughout the course. The lateral occipital area (Figure 2) showed a similar emergence of responses to tactile Braille words after the course as well.”

As to what causes the drop of activation to control stimuli, we believe that the pattern of activation in SI and SII is very informative here. In SI, but most vividly in SII, there is a general drop in activation to all tactile stimuli after the course of 15 and 31 percent, respectively (Figure 5).

Author response image 1.SI and SII activity for tactile stimuli before and after the course.**DOI:**
http://dx.doi.org/10.7554/eLife.10762.014

This most likely reflects more efficient tactile processing. It is plausible that this decrease in the somatosensory cortex also led to a general decrease of somatosensory response in the higher visual cortex (Figure 6). As the below figure explains, our interpretation is that the magnitude of the final response is most likely a result of two processes – a general decrease of activation for all somatosensory stimuli (both tactile Braille words and control), and a specific increase of activation for Braille words.

Author response image 2.Possible contributions to the decrease of activity for tactile stimuli.Two contributions to interaction for tactile stimuli (example for Visual Word Form Area, based on Figure 2).**DOI:**
http://dx.doi.org/10.7554/eLife.10762.015

A second factor contributing to increased deactivation to control stimuli after the course might be a disengagement of attention from control stimuli. Before the course, our subjects could not distinguish Braille words from control stimuli. After the course, Braille words became clearly distinguishable to them. The subjects could therefore disengage their attention rapidly from tactile control stimuli. One can see this very significant (p<0.001) disengagement in ROI analysis of the IPS (Figure 2). Disengagement of IPS for written word stimuli is a known hallmark of reading activations, that appears quite early during the course of reading development.

To reflect the above in the new version of the manuscript, we have made the following change to the description of the ROI results in the somatosensory cortices and the IPS:

“Those activation drops in the somatosensory cortices and in IPS most likely led to the activation drop for control tactile stimuli observed in the VWFA and LO (Figure 2).”

*2) TMS study* –

although the lateral occipital area was chosen here as a control site, doesn't the FMRI data (Figure 2) suggest it shows a similar effect to the VWFA? How should this be reconciled?

The Lateral Occipital area (LOA) plays a crucial role in object recognition (Grill-Spector et al., 2001). It is also activated in various word recognition tasks (Duncan et al., 2009; Wright et al., 2008). However, the exact role of LOA in reading remains unknown. Its lesion seems not to affect reading itself (Philipose et al., 2007). Milner et al., (1991), already reported a case study of a patient with left lateral occipital lesion who suffered from visual agnosia with reading functions relatively unimpaired. On the other hand, the LOA is known to be activated when advanced readers start reading a novel script. For example, French native speakers activated bilateral LOA when observing Chinese characters – stimuli that were not familiar to them (Szwed et al., 2014)(Figure 7).

Author response image 3.LOA activates when French, Chinese-naive native speakers see Chinese characters.Source [64].**DOI:**
http://dx.doi.org/10.7554/eLife.10762.016

Based on those observations and a large body of existing results, including the TMS work by Zangaladze, Epstein, Grafton, and Sathian, (1999) and Merabet et al., (2004) discussed in the eighth paragraph of the Discussion, a plausible explanation is that the LOA is responsible for analysing object features, including tactile object features. However, it is not critical for visual and Braille word recognition.

The process of reading itself engages a more ventral area – VWFA, widely connected with left-hemispheric language areas (Bouhali et al., 2014). This is why TMS application to LOA does not influence reading accuracy, whereas inhibition of the VWFA does (Duncan, Pattamadilok, and Devlin, 2010, and our own results presented here). This is because, we believe, the LOA activity is not critical for visual and Braille word recognition. This interpretation is consistent with the fact that the main effect in the LOA was only in the ROI analysis. When we correlated tactile Braille final reading speed with the whole-brain activity to tactile Braille words vs. control after the course we found only a small cluster of 41 voxels (p=0.005 voxel-wise, uncorrected) located in the right LOA (MNI 51 -55 12). The increase of activity in both LOA and VWFA for tactile reading suggest that visual and tactile reading share similar neural correlates along the ventral visual stream. However, the exact function of LOA in reading seems to be more accessory than critical.

We have added following paragraph in the Discussion:

“ROI analysis (Figure 2) revealed that lateral occipital area (LOA) presented a pattern of activity increase to tactile words due to the course similar to the VWFA. […] However, the exact function of LOA in reading seems to be more accessory than critical.”

3) In the Discussion, it should be explained why several experiments failed to find similar reorganization in sighted subjects, even after extensive training (Kupers et al., 2006; Ptito et al., 2005).

We believe that the main cause for different results between our study and the study of Kupers et al., 2006 and Ptito et al., 2005 was the length and the character of the training. In both papers subjects were trained in a simple tactile discrimination task. Ptito and colleagues trained their subjects for seven days, Kupers et al. – for two days. Our experiment on the other hand was longer (9 months) and used complex, entire words as stimuli. Indeed, experiments that study learning-related plasticity at multiple time points (Lövdén et al., 2013) suggest that at the initial stage of Braille learning, the somatosensory cortex might have increased its response to Braille words. Then, as the effects of early sensory learning consolidated in somatosensory cortex and the cortical focus of learning shifted elsewhere, in our case the ventral visual stream. Because the above-mentioned studies used simple stimuli and short learning periods, they captured only the initial reorganization in the somatosensory system.

The following lines were modified and added in the Discussion:

“Cortical reorganization that crosses the sensory boundaries is prominent in blind and deaf humans (Hirsch et al., 2015; Pavani and Roder, 2012; Sadato et al., 2002) and in congenitally deaf cats (Lomber et al., 2011). […] Then, as the effects of early sensory learning consolidated in the somatosensory cortex, the cortical focus of learning shifted elsewhere, in our case the ventral visual stream.

*4) Figure 1 shows a correlation between post-training activity during tactile Braille reading and Braille reading speed. How many subjects had a reading speed of 0 WPM? Is the spread of reading speed suitable for a linear regression analysis?*

There were 2 subjects who read 0 words per minute at the end of the Braille course. The distribution of reading speed was normal (Shapiro-Wilk (29)= 0.94, p = 0.123, Figure 1), with the mean =6.21, SD=3.94, median=5, range 0-17).

Author response image 4.The distribution of final Braille reading speed (words per minute, WPM).**DOI:**
http://dx.doi.org/10.7554/eLife.10762.017

The subject who read 17 words per minute was an outlier (Figure 1). In order to check whether the final outcome of the model was not due to the effect of this outlier, we recalculated the model excluding this subject. At p=0.005 voxelwise we found a cluster of 50 voxels including left middle occipital gyrus (z=3.45, BA19, MNI -42 -85 -1) and left inferior occipital gyrus (z=2.92, BA19, MNI -45 -73 -13, Figure 2).

Author response image 5.Tactile Braille modulated by reading speed after course (outlier excluded).Voxel-wise p=0.005, k=50.**DOI:**
http://dx.doi.org/10.7554/eLife.10762.018

We therefore demonstrate that while the outlier influenced cluster size and Z value, exclusion of the outlier did not change the localization of the effect: the only neural correlate of Braille reading speed was found in the left visual cortex.

We thus conclude that the spread of final Braille reading speed was suitable to perform regression analysis. As the exclusion of the outlier did not change dramatically the overall message of the SPM regression analysis, we decided not to exclude this data from the final model presented in the paper. We hope that this additional information answers the reviewers’ questions.

The following information was added to the Appendix 1.1:

“The distribution of final reading speed was normal (Shapiro-Wilk (29)=0.94, p = 0.123), with the mean=6.21, SD=3.94, median=5, mode=4, range 0-17. There were 2 subjects who read 0 words per minute at the end of the Braille course.”

*5) For the correlation between resting connectivity strength and change in reading speed over the preceding month (Figure 3)* –

how many behavioural variables were tested for correlation here and should some correction be made for multiple comparisons? Why change over the past month rather than some other time interval? And why change in reading speed rather than simply reading speed (when the latter was considered for Figure 1 for example)?

We agree that the large number of behavioral sessions in our study pose a risk of improper use of statistical methods, which may lead to biased results. However, we believe that correlation presented in Figure 3 is valid and robust.

Previous studies show that training can lead to rapid change in resting-state functional connectivity pattern. For example, Lewis et al. (2009) demonstrated that 2 to 9 days of perceptual visual training modifies functional connectivity between the visual cortex and frontal regions. Urner et al. (2013) showed that even one training session in motion dots coherence detection task changes functional connectivity between hippocampus and striatum, and that this change is preserved a day after the training. Rapid changes in resting-state functional networks can be also observed following complex tasks learning. Voss et al., (2012) show that 20 hours of videogame training leads to changes in functional connectivity of fronto-executive network.

Based on the above-mentioned studies, we expected that final resting-state functional connectivity would mostly reflect subjects’ intensity of training in last days of the Braille course. In our study, this can be quantified as a change in Braille reading speed over last month of the course. This behavioral measure was thus used for correlation. *Given our a priori hypothesis, we believe that applying correction for multiple comparisons is not necessary in this case.* Note that absolute Braille reading speed is more likely to reflect regularity of training across the whole course. Such a measure is well suited for correlation with the task-based fMRI results (Figure 1), but it is probably not optimal in the case of resting-state fMRI analysis.

Having this in mind, we want to stress that correlation presented in Figure 3 remains significant even when conservative correction for multiple comparisons is applied. In our study, behavioral data were collected in 6 testing sessions (beginning of the course, 5^th^ month, 6^th^ month, 7^th^ month, 8th month and end of the course). Thus, correlating functional connectivity measure with all meaningful time intervals that can be formed (end of the course – 8^th^ month, end of the course – 7^th^ month, end of the course – 6^th^ month, end of the course – 5^th^ month and end of the course – beginning of the course), as well as with absolute final reading speed, would form 6 comparisons. Applying conservative Bonferroni correction to account for these comparisons would yield statistical threshold of p = 0.008 uncorrected, equal to p = 0.05 corrected for 6 comparisons. Correlation that we demonstrated in Figure 3 is significant at statistical threshold of p = 0.007 uncorrected, being equal to p = 0.042 Bonferroni-corrected for 6 multiple comparisons. *Thus, the correlation presented in Figure 3 can be robustly observed even when we apply correction that accounts for a random search in all meaningful behavioral measures*.

In summary, our choice of behavioral measure was based on a priori hypothesis and previous studies. We believe that applying correction for multiple comparisons is not necessary. However, applying a conservative correction would not change our results, which confirms that correlation presented in Figure 3 is robust. We are ready to report corrected significance value if reviewers ask us to do this.

To make these points clearer, we added following lines to the manuscript (Materials and methods):

“Previous studies show that training can lead to rapid change in resting-state functional connectivity pattern (e.g., Lewis et al., 2009, Urner et al., 2013; Voss et al., 2011). […] However, the same result was obtained even when correction for multiple comparisons was applied to account for multiple behavioural sessions.”

6) In the subsection “Representation Similarity Analysis”: The claim that the correlations of Braille conditions with the visual word condition were weaker than those between the two Braille conditions should be substantiated by appropriate statistical tests. The tests reported in Section 1.6 of the Appendix do not seem to address this issue.

We believe that the reviewers had in mind the Section 1.6 of the Appendix (Representation Similarity Analysis: p values). We agree thatthissection was perhaps too short and failed to adequately convey the statistical tests we actually did.

In that section, to compare similarities between examined scripts we calculated correlation coefficients of pairwise correlations between them (visual words x visual Braille, visual Braille x tactile Braille and tactile Braille x visual Braille). Correlation coefficients were then Fisher r to z transformed and compared using paired t-tests.

Thus, we performed paired t-tests of the correlation coefficients of following pairs:

1) Tactile Braille x visual Braille and visual Braille x visual words, t(28)=-2.94, p=0.007;

2) Tactile Braille x visual Braille and tactile Braille x visual words, t(28)=-5.21, p<0.001;

3) Tactile Braille x visual words and visual Braille x visual words, t(28)=-2.45, p=0.021.

Using t-tests in assessment of differences between neural patterns across different regions or different cognitive functions is a standard procedure (see e.g. Bannert and Bartels, 2013; Chikazoe, Lee, Kriegeskorte, and Anderson, 2014).

We believe that reviewers’ remark considered pairwise comparisons of mean correlation coefficients using paired T-tests. We understand this remark, as performing series of t-test without the correction for multiple comparisons can lead to increase probability of type I error. We therefore recalculated the data as following:

In order to address the relation between neural patterns of three examined scripts, we calculated a one-way repeated measures ANOVA with Fisher r-to-z transformed correlation coefficients (z(r)) as dependent variable and the factor of script pairs (3 levels: tactile Braille and visual Braille, tactile Braille and visual words, and visual Braille and visual words). The main effect of scripts’ pairs was highly significant (F(2,56)=14.53, p<0.001). Simple effects of scripts’ pairs were assessed using post-hoc tests with Bonferroni correction. The correlation between the two Braille conditions was significantly higher than the correlation between tactile Braille and visual Words (p<0.001) and visual Braille and visual Words (p=0.020). The correlation between visual Braille and visual words was higher than the correlation between tactile Braille and visual Words, however this result was only marginally significant (p=0.063).

To sum up, *even when conservative Bonferroni correction for multiple comparisons was applied, the correlation between two Braille scripts remained significantly stronger than correlations between visual and Braille conditions*. The correction affected however the level of statistical significance of the difference between correlations of visual words with Braille conditions. Corrected p values together with modified Methods section were inserted in the manuscript. We hope that this modification of statistical methods meets reviewers’ concerns.

The following lines were added to the main text (Materials and methods):

“The resulting correlation coefficients for each condition pair were Fisher r-to-z transformed (z(r)) and compared on a group level in one-way repeated measures ANOVA with a factor of script pairs (3 levels: tactile Braille and visual Braille, tactile Braille and visual words, and visual Braille and visual words). Simple effects were analyzed using post-hoc tests with Bonferroni correction.”

And to Appendix 1.6:

“The main effect of scripts’ pairs was highly significant (F(2,56)=14.53, p<0.001). Simple effects of scripts’ pairs were assessed using post-hoc tests with Bonferroni correction. The correlation between the two Braille conditions was significantly higher than the correlation between tactile Braille and visual Words (p<0.001) and visual Braille and visual Words (p=0.020). The correlation between visual Braille and visual words was higher than the correlation between tactile Braille and visual Words, however this result was only marginally significant (p=0.063).”

7) The authors used a rather rapid rate of rTMS (10 Hz). Although this has indeed been used to suppress activity as in the prior study of rTMS over the VWFA, this rate is sometimes considered facilitatory. Could the authors comment on this?

We agree that some studies suggest that low-frequency (i.e., ≤ 1Hz) stimulation decreases cortical excitability while high-frequency (i.e., ≥ 1Hz) stimulation increases cortical excitability. This classification of TMS frequencies can be clearly illustrated in the motor system where low-frequency rTMS delivered to primary motor cortex reduces the amplitude of motor evoke potential (MEP) while high-frequency rTMS enhances MEP amplitude (Berardelli et al., 1999; Chen et al., 1997; Jennum et al., 1995; Maeda et al., 2000; Pascual-Leone et al., 1994; Rossi et al., 2000). However, it is less clear that these findings generalize to areas outside the motor cortex. For instance, to induce speech arrest (i.e., disruption of speech production), rTMS at rather high frequencies (4-32 Hz) has been used over the left prefrontal cortex (Epstein et al., 1996; Jennum et al., 1994; Pascual-Leone et al., 1991). Similarly, the majority of studies using either high- or low-frequency rTMS to areas involved in cognitive processes showed disruptive, rather than facilitatory, effects on behavioural measures such as reaction times or accuracy (Sliwinska et al., 2014; Gough et al., 2005; Hartwigsen, et al., 2010; Pitcher et al., 2007; Pobric et al., 2010; Romero et al., 2006; Whitney et al., 2010). Consequently, these results demonstrate that it may be somewhat simplistic to classify a stimulation protocol as inhibitory or facilitatory based solely on the frequency of stimulation.

We admit that choosing a specific frequency of stimulation is challenging because different values are likely to work equally well. There are, however, some heuristic guidelines that helped us to constrain the choice. Low-frequency rTMS is used in off-line TMS experiments where long-lasting stimulation is believed to have an inhibitory after-effect lasting from 30-60 min, depending on the duration and intensity of the stimulation (Ridding and Rothwell, 2007). On-line experiments, such as ours, tend to use high-frequency rTMS to produce short-lasting inhibition of cognitive processes. Many studies have used 10 Hz stimulation during task performance to slow reaction times (Göbel et al., 2001) and/or induce errors (Hartwigsen et al., 2010). In fact, as mentioned in the Materials and methods section (subsection “TMS protocol”), we chose the specific paradigm that used rTMS at frequency of 10 Hz for 500 ms since it has proven to be very effective and robust for producing virtual lesions across different cortical areas (Bjoertomt et al., 2002; Duncan et al., 2010; Göbel et al., 2001; Hartwigsen, Price, et al., 2010; Lavidor and Walsh, 2003; Pitcher et al., 2007; Rushworth, Ellison, and Walsh, 2001), not only the Visual Word Form Area (Duncan et al., 2010).

8) In the Introduction, considerable attention is devoted to the idea that the behavioral relevance of the cross-modal recruitment of visual cortex demonstrated in the present study is highly novel. Without detracting from the novelty of the study, it would be appropriate to acknowledge earlier studies that did demonstrate such behavioral relevance, e.g., Zangaladze et al., Nature, 401: 587-590, 1999; Merabet et al., Neuron, 42: 173-179, 2004.

We agree with the reviewers that those papers add important information about the state of the art concerning visual cortex engagement in non-visual task. The pioneer work by Zangaladze et al., 1999, showed that TMS over occipital cortex interferes with tactile grating orientation task in the sighted, proved visual cortex engagement in tactile tasks. Merabet et al. (2004), demonstrated double dissociation between the engagement of somatosensory and visual cortex in tactile discrimination task in healthy sighted subjects. Low frequency rTMS applied to somatosensory cortex affected the judgment of roughness/texture, but not the distance between stimuli. rTMS to visual cortex yielded opposite results, disrupting the judgment of distance, but not roughness. Therefore, we now:

Mention the two (very relevant) citations in Introduction (second paragraph);

We removed the sentence: “none of them demonstrated that such cortical changes are behaviorally relevant”(Introduction, second paragraph)

Instead, we write: “the behavioural relevance of these cortical mechanisms remains unclear, especially for complex stimuli”.

We then discuss the two above-mentioned studies in the Discussion section:

“The behavioral relevance of such activations, however, was demonstrated only for simple tactile stimuli: grating orientation (Zangaladze et al., 1999) and distance judgement (Merabet et al., 2004).”